# Investigating three patterns of new particles growing to the size of cloud condensation nuclei in Beijing's urban atmosphere

Liya Ma[1#], Yujiao Zhu[1, 2#]*, Mei Zheng[3], Yele Sun[4], Lei Huang[1], Xiaohuan Liu[1], Yang Gao[1], Yanjie Shen[1], Huiwang Gao[1, 5] and Xiaohong Yao[1, 5]*

[1]Lab of Marine Environmental Science and Ecology, Ministry of Education, Ocean University of China, Qingdao 266100, China

[2]Environment Research Institute, Shandong University, Qingdao, Shandong 266237, China

[3]State Key Joint Laboratory for Environmental Simulation and Pollution Control, College of Environmental Sciences and Engineering, Peking University, Beijing 100871, China

[4]State Key Laboratory of Atmospheric Boundary Layer Physics and Atmospheric Chemistry, Institute of Atmospheric Physics, Chinese Academy of Sciences, Beijing 100029, China

[5]Laboratory for Marine Ecology and Environmental Sciences, Qingdao National Laboratory for Marine Science and Technology, Qingdao, China

[#]*Equally contributed to this work; Correspondence to*: Xiaohong Yao and Yujiao Zhu (*[\*xhyao@ouc.edu.cn](mailto:xhyao@ouc.edu.cn), [zhuyujiao@sdu.edu.cn](mailto:zhuyujiao@sdu.edu.cn))*

**Abstract.** The growth of newly formed particles with diameters from ~10 nm to larger sizes was investigated in Beijing's urban atmosphere during December 10–23, 2011, April 12–27, 2012, and June–August 2014. In 11 out of 27 new particle formation (NPF) events during June–August, the maximum geometric median diameter ($D_{pgmax}$) of newly formed particles exceeded 75 nm, and the grown new particles may contribute to the population of cloud condensation nuclei. In contrast, no apparent growth in new particles with $D_{pgmax}<20$ nm was observed in all of the events in December, in approximately half of the NPF events in April and only 2 events during June-August. New particles observed in the latter NPF events were too small to be activated as cloud condensation nuclei. Apparent new particle growth with $D_{pgmax}\leq50$ nm was observed in the remaining 18 NPF events. The 11 NPF events during June–August with $D_{pgmax}$ exceeding 75 nm were analyzed in detail. The particle growth patterns can be clearly classified into three types: one-stage growth, and two-stage growth-A and growth-B. The one-stage growth pattern is characterized by a continuous increase in $D_{pg}$ with $D_{pgmax} \geq 80$ nm (4 out of 11 NPF events), and the two-stage growth-A and growth-B patterns are characterized by no apparent growth and shrinkage of particles, respectively, in the middle 2–4 h of the growth period (7 out of 11 NPF events). Combining the observations of gaseous pollutants and measured (or modeled) concentrations of particulate chemical species, the three growth patterns were discussed in terms of the spatial heterogeneity of NPF, formation of secondary aerosols, and evaporation of semi-volatile particulates. Secondary organic species and $NH_4NO_3$ were argued to be two major contributors to the growth of new particles, but $NH_4NO_3$ likely contributed to growth only in the late afternoon and/or at nighttime.

## 1 Introduction

Atmospheric aerosol particles can be derived either from primary emissions, including various natural and anthropogenic

sources, or from secondary sources (Yao et al., 2005; Sabaliauskas et al., 2012; Vu et al., 2015; Seinfeld and Pandis, 2016; Quan et al., 2017; Zhu et al., 2019). Secondary sources are mainly related to atmospheric nucleation, followed by the growth of newly formed particles from ~1 nm to larger sizes; this phenomenon is conventionally referred to as a new particle formation (NPF) event (Kulmala et al., 2004; Kerminen et al., 2018). In recent decades, numerous studies have been conducted on NPF,

including field measurements in various atmospheres, laboratory studies on nucleation and initial growth in newly formed particles, regional-scale modeling of NPF and its impacts on climate, and development of new techniques for analyzing the chemical components of nanoparticles and their gaseous precursors. Building on the results of these studies, many review papers have summarized the state-of-the-art progress and noted challenges for future studies (Kulmala et al., 2004, 2012, 2016; Kulmala and Kerminen, 2008; Zhang et al., 2012, 2015; Kerminen et al., 2018; Chu et al., 2019; Lee et al., 2019).

Several studies have investigated the potential climate impacts of NPF events, for example, 10% to 60% of NPF events have been reported to yield an appreciable contribution to cloud condensation nuclei (CCN) (Kuang et al., 2009; Asmi et al., 2011; Laakso et al., 2013; Yu et al., 2014; Rose et al., 2017; Kerminen et al., 2018). Modeling studies have also proposed that approximately 50% of the CCN population is attributable to NPF events in the troposphere (Yu and Luo, 2009; Yu et al., 2014; Gordon et al., 2017). Nevertheless, reported observations have also shown that newly formed particles with diameters less than

40–50 nm can be activated as CCN only under high supersaturation (SS), such as >0.6% (Li et al., 2015; Ma et al., 2016). When newly formed particles grow with the geometric median diameter to larger than 70 nm, they significantly contribute to the CCN population at SS≤0.2% (Wiedensohler et al., 2009; Yue et al., 2011; Li et al., 2015; Ma et al., 2016; Zhu et al., 2019). In addition, field observations have also shown that in most NPF events, the maximum geometric median diameter ($D_{pgmax}$) of newly grown particles is less than 40–50 nm before new particle signals drop to a negligible level (Zhu et al., 2014, Liu et al.,

2014; Man et al., 2015; Zhu et al., 2017; Yu et al., 2019). Thus, it is important to characterize NPF events, based on the $D_{pgmax}$ of grown new particles, and to explore the chemicals driving the growth of newly formed particles with $D_{pgmax}$ greater than 70 nm.

With distinctive particle growth patterns being widely reported, Beijing is an ideal area for studying the growth of newly formed particles (Wehner et al., 2004; Wu et al., 2007, 2016; Wiedensohler et al., 2009; Yue et al., 2010; Matsui et al., 2011;

Wang et al., 2013; Guo et al., 2014, 2020; Du et al., 2017; Zhu et al., 2017; Brean et al., 2019; Chen et al., 2019). For instance, as the first study of NPF events in Beijing, Wehner et al. (2004) reported a small growth rate (~1 nm h$^{-1}$) of newly formed particles during 25 days from March 05 to April 18, 2004. Such small growth rates are unlikely to facilitate the growth of particles to reach CCN sizes prior to removal from ambient air because of the large coagulation loss in Beijing (Kulmala and Kerminen, 2008; Kulmala et al., 2016; Chu et al., 2019; Guo et al., 2020). Similar to this finding, no apparent growth of newly

formed particles with the $D_{pgmax}$ of approximately 10 nm always occurred in December 2011 at the same campus in Beijing (Zhu et al., 2017). In contrast, the growth of newly formed particles to CCN size and even larger has also been observed in

Beijing (Wu et al., 2007; Wiedensohler et al., 2009; Yue et al., 2010; Wang et al., 2013; Guo et al., 2014; Wu et al., 2016). The patterns of particle growth have not been well characterized. For example, the occurrence frequency of NPF events in which newly formed particles can grow to CCN size is virtually unknown. The same can be said to season trends in the occurrence frequency. Nevertheless, sulfuric acid and/or organic vapors have been proposed to drive particle growth in different NPF events (Wiedensohler et al., 2009; Yue et al., 2010; Wu et al., 2016). Recently, the formation of $NH_4NO_3$ has been proposed as a driver of the rapid growth of newly formed particles in field studies and chamber experiments (Zhu et al., 2104; Man et al., 2015; Wang et al., 2020). The role of $NH_4NO_3$ in the growth of newly formed particles in Beijing remains poorly understood. The contributions of sulfuric acid, organic vapors and $NH_4NO_3$ to the growth of newly formed particles at different sizes are also non-existent. Matsui et al. (2011) and Chen et al. (2019) simulated NPF and the growth of newly formed particles based on observations, but the modeling results were explained with large uncertainties.

In this study, we investigated NPF events in Beijing using observational data from three campaigns. We characterized NPF events according to the observed $D_{pgmax}$ of newly formed particles and then focused on analyzing the growth patterns of newly formed particles with diameter from ~10 nm to a larger size, paying particular attention to NPF events in which the $D_{pgmax}$ exceeded 70 nm. We combined observations of gaseous pollutants and observed (or modeled) concentrations of organic matter (or secondary organic aerosol, SOA), $NO_3^-$, and $NH_4^+$, to identify the chemicals driving the growth of new particles, for example, varying major contributors in different growth periods. The survival probability of newly formed particles, which can grow over 50 nm or 70 nm (two CCN threshold sizes under different SS), was also estimated. Our study provides new insight into the growth of newly formed particles to larger sizes, as required for these particles to be activated as CCN at normal SS in the atmosphere.

## 2 Methods

### 2.1 Sampling periods, sites, and instruments

Two sampling sites were adopted to measure the particle number concentration spectra in Beijing. One is a rooftop site on the roof of an academic building within the campus of Peking University (39.99°N, 116.31°E, ~20 m above ground level); the other is a street site along a road located approximately 200 m from the rooftop site (Fig. 1). At the rooftop site, measurements were taken on December 16–23, 2011, April 12-27, 2012, and June 1–August 31, 2014. At the street site, measurements were taken on December 10–23, 2011 and April 18–27, 2012. The concentrations of atmospheric particles were measured using a fast mobility particle sizer (FMPS, TSI Model 3091) downstream of a dryer (TSI, 3091) at a one-second time resolution in each measurement campaign. The FMPS was a paralleling particle sizer and reported number size distributions of aerosol particles from 5.6 nm to 560 nm. In this study, the empirical correction procedure for FMPS size distribution data proposed by Zimmerman (2015) was used for correction. The scaling-down coefficient of the total particle number concentration measured by the FMPS (1.28) was obtained through a correlation analysis of side-by-side measurements

made by the FMPS and a condensation particle counter (CPC). A $SO_2$ analyzer (Thermo Model 43i), an $O_3$ analyzer (Thermo Model 49i), a $NO_X$ analyzer (Thermo Model 42i), and a meteorological monitoring system were operated at a one-minute resolution to obtain real-time observational data of gases and meteorological parameters on the rooftop site close to the FMPS in 2011, 2012, and before July 10, 2014. During the other observational periods (July 11–August 31, 2014), the mixing ratios of air pollutants at a one-hour resolution and meteorological data at a three-hour resolution were taken from the Wanliu Air Quality Monitoring Station in Haidian district (39.99°N, 116.32°E, http://106.37.208.233:20035/) and the Beijing 54511 station (39.95°N, 116.30°E, https://rp5.ru/), respectively. The concentrations of oxygenated organic aerosols (OOA) and inorganic species including $NO_3^-$, $SO_4^{2-}$, and $NH_4^+$ in $PM_{1.0}$, during the period from June 3–July 11, 2014, previously reported by Xu et al. (2017), were also used to facilitate the analysis. The data were measured using a High-Resolution Time-of-Flight AMS (HR-ToF-AMS) at 10 min resolution. The chemical composition of $PM_{1.0}$ measured by AMS has been widely used to interpret NPF events in the literature (Wiedensohler et al., 2009; Zhang et al., 2014; Man et al., 2015; Du et al., 2017; Rodelas et al., 2019; Kanawade et al., 2020) and was also used in this study. Low loadings of particulate chemical species in nanometer size ranges do not facilitate accurate measurement of their concentrations therein. However, the chemical composition of nanometer particles may differ from those of $PM_{1.0}$ (Ehn et al., 2014; Wu et al., 2016). Moreover, the sampling site of the HR-ToF-AMS was located at the Tower branch of the Institute of Atmospheric Physics in Beijing, China (39.98°N, 116.38°E), and approximately 8 km away from Peking University. During NPF events at wind speeds of 4–6 m s$^{-1}$, a half-hour delay may occur for air parcels sweeping from one site with the FMPS deployed to another site with the AMS deployed. For NPF events with durations over several hours, the events were expected to occur regionally (Kerminen et al., 2018; Chu et al., 2019). Thus, it is reasonable to interpret the cumulative growth of newly formed particles within several hours, measured by the FMPS, by using the net simultaneous change in concentrations of chemical species, measured by the AMS. Additionally, He et al. (2001) reported that the chemical composition of $PM_{2.5}$ was reasonably homogeneous in the two sampling site zones.

## 2.2 Computational methods

NPF events were identified according to the definition by Dal Maso et al. (2005), and only NPF events with durations over one hour were analyzed in this study. The local standard time was used to describe the NPF events in this study. In each NPF event, the net maximum increase in the nucleation mode particle number concentration (NMINP) was calculated according to Zhu et al. (2017). The nucleation mode was defined from 8 to 20 nm in this study.

$$NMINP = N_{8-20\ nm}\ (t_1) - N_{8-20\ nm}\ (t_0) \tag{1}$$

$N_{8-20\ nm}$ represents the sum of particle number concentrations with diameters from 8 nm to 20 nm; $t_0$ and $t_1$ represent the time of the initial observation of an NPF event and the time at which $N_{8-20\ nm}$ arrives at the maximum value, respectively. Note that a few spikes of several minutes were occasionally observed and were excluded to calculate NMINP because they may reflect primary particles from localized sources (Liu et al., 2014; Zhu et al., 2017).

The growth rate (GR) and shrinkage rate (SR) of new particles are determined by the slope of the fitted geometric median diameter of new particles ($D_{pg}$) over time (Whitby et al., 1978; Yao et al., 2010; Zhu et al., 2014; Man et al., 2015). In an NPF event or in each growth period of one NPF event, the maximum value of $D_{pg}$ is defined as $D_{pgmax}$. Again, a few occasional spikes of several minutes were excluded to calculate GR, SR, and $D_{pgmax}$ (Liu et al., 2014; Zhu et al., 2017).

5        Lu et al. (2019) recently developed an equation to estimate the gaseous sulfuric acid concentration in Beijing. The equation is expressed as follows:

$$[H_2SO_4] = 280.05 \cdot UVB^{0.14} \cdot [SO_2]^{0.40} \tag{2}$$

The units for $[H_2SO_4]$ and $[SO_2]$ are molecule cm$^{-3}$, and the unit for UVB (ultraviolet B) is W m$^{-2}$. The reported error was within 20% for the calculated concentrations against the observations in Beijing (Lu et al., 2019). UVB occupies 5% of the ultraviolet radiation that reaches the Earth's surface (https://en.wikipedia.org/wiki/Ultraviolet#cite_note-Skin_Cancer_Foundation-23). Thus, UVB values were obtained by multiplying the downward ultraviolet radiation at the surface by 5% in this study, and the ultraviolet radiation data were downloaded from the Climate Data Store (https://cds.climate.copernicus.eu/). The contribution of sulfuric acid vapor to particle growth was calculated based on the method reported by Kulmala et al. (2001) and Nieminen et al. (2010).

$$R = ([H_2SO_4]_{avg}/C) \times 100\% \tag{3}$$

where $[H_2SO_4]_{avg}$ is the average concentration of $H_2SO_4$ vapor during the particle growth period, and C is the total concentration of condensable vapor for the particle growth. Here, the surface vapor pressure of the condensable material is assumed to be zero, and $C$ can be calculated based on the equation of Kulmala et al. (2001).

$$C = \rho\{\frac{D_{pg2}^2 - D_{pg1}^2}{2} + [\frac{4}{3a} - 0.623]\lambda(D_{pg2} - D_{pg1}) + 0.623\lambda^2 \ln\frac{\lambda + D_{pg2}}{\lambda + D_{pg1}}\}/\Delta t D m \tag{4}$$

$\rho$ is the particle density in g cm$^{-3}$, $a$ is the mass accommodation coefficient (i.e., sticking probability), $\lambda$ is the mean free path in nm, $\Delta t$ (s) is the time during particle growth from $D_{pg1}$ to $D_{pg2}$, $D$ (cm$^2$ s$^{-1}$) is the diffusion coefficient of the condensing vapor, and $m$ is the molecular mass of the condensable vapor in g mol$^{-1}$.

       To calculate the survival probability (SP) of grown new particles beyond 50 nm or 70 nm, temporal variations in $N_{50-200}$ $_{nm}$ and $N_{70-200\,nm}$ were plotted, e.g., Fig. S2a-b showed the temporal results on 25 August 2014. With increasing of the height of planetary boundary layer (PBL, Fig. S2d), the stable minimum $N_{50-200\,nm}$ or $N_{70-200\,nm}$ can be clearly identified approximately 2-3 hours after the NPF event to be observed. The minimum $N_{50-200\,nm}$ ($N_{70-200\,nm}$) remained constant for approximately two (four) hours, even though the height of PBL increased continuously. The new particle signal likely mixed well within the whole PBL. The stable maximum $N_{50-200\,nm}$ or $N_{70-200\,nm}$ can be also clearly identified approximately 11-13 hours later, when the height of PBL has been lowered down a lot from the maximum. The change in height of PBL had no detectable influence on the maximum $N_{50-200\,nm}$ or $N_{70-200\,nm}$ because the growth of >30 nm new particles to a larger size probably occurred within

 The difference between the two values was thereby used to estimate the net maximum number concentration of grown new particles beyond 50 nm or 70 nm, i.e., $N_{50\text{-}200\,nm}$ (*net*) and $N_{70\text{-}200\,nm}$ (*net*). Thus, $SP_{50\,nm}$ and $SP_{70\,nm}$ were estimated as:

$$SP_{50\,nm} = \frac{N_{50-200\,nm}(net)}{NMINP} \times 100\% \tag{5}$$

$$SP_{70\,nm} = \frac{N_{70-200\,nm}(net)}{NMINP} \times 100\% \tag{6}$$

Note that the observed number concentrations of newly grown particles with a larger size sometimes exceeded those with a smaller size under the condition of spatial heterogeneity of NPF. In these cases, that is, NPF events occurring on June 23, and August 12 and 15, SP was not calculated.

Followed Equation (4) proposed by Kulmala et al., (2001), in which $D_{pg1}$ and $D_{pg2}$ are used to calculate C, the amount of chemical species required to grow new particles from $D_{pg1}$ to $D_{pg2}$ ($Mass_{requried}$) is approximately estimated as follows:

$$Mass_{requried} = 4/3\pi\,[(D_{pg2}/2)^3 - (D_{pg1}/2)^3] * N * \rho \tag{7}$$

$\rho$ is the density, which is assumed as 1.5 µg m$^{-3}$ for OOA (or SOA) and 1.7 µg m$^{-3}$ for $NH_4NO_3$, respectively. Considering that the particle number concentration may decrease because of the dry deposition, diffusion and dilution effects, and particle coagulation, N represents the integral value of new particle number concentrations with the geometric median diameter of new particles from $D_{pg2\text{-}3\sigma}$ to $D_{pg2\text{+}3\sigma}$. The approximate value may overestimate the required amount because particle–particle coagulation has not been deducted.

## 2.3 Model description

The U.S. EPA Community Multiscale Air Quality Model (CMAQ version5.0.2, Byun and Schere, 2006) was applied to simulate inorganic ions such as $NO_3^-$, $SO_4^{2-}$, $NH_4^+$, and SOA in $PM_{2.5}$ over East Asia. Fig. S1 shows the nested domains, with the 36 km domain 1 (d01) and the 12 km domain 2 (d02) over eastern China and China adjacent seas. The vertical resolution includes 14 logarithmic structure layers from the surface to the tropopause, with the first model layer height of 36 m above the ground level. Meteorological fields were generated by the Weather Research and Forecasting (WRF) Model (v3.7). The initial and boundary conditions were obtained from the National Center for Environmental Prediction (NCEP) FNL (Final) Operational Global Analysis datasets (http://rda.ucar.edu/datasets/ds083.2). The major physics options included the Lin microphysics scheme, RRTM long-wave radiation scheme, Goddard short wave scheme, Monin-Obukhov surface-layer scheme, thermal diffusion land-surface scheme, and YSU land-surface scheme. The WRF hourly output files were processed using the Meteorology-Chemistry Interface Processor (MCIP v4.3). In CMAQ, the CB05tucl chemical mechanism module coupled with AERO6 was used to simulate the concentrations of gases and aerosols. Initial conditions (ICONs) and boundary conditions (BCONs) of pollutants in d01 were generated using the results from a global chemistry model of GEOS-Chem,

while ICONs and BCONs for d02 were obtained from the results of d01. The Multi-resolution Emission Inventory for China (MEIC) in 2014, developed by Tsinghua University (http://www.meicmodel.org/), combined with BVOC emissions generated from Gases and Aerosols from Nature (MEGAN v 2.0.4, Guenther et al., 2006) was used in this study. A spin-up time of five days was used to minimize the influence of ICONs.

Liu et al. (2010 a, b), Qi et al. (2018), and Zhang et al. (2019) reported the validation of the CMAQ application in China, in which good agreements between the simulated and measured concentrations of particulate components were generally obtained. During the study period, the model results of $SO_4^{2-}$, $NH_4^+$ and $NO_3^-$ met the benchmark criteria of the above three species (US-EPA, 2007), with correlations between modeled and measured values larger than 0.61 (Table S1). The modeled concentrations of $NH_4^+$ reasonably agree with the observations with a normalized mean bias (NMB) of 6%. The NMB slightly increased up to 12% for the modeled concentrations of $SO_4^{2-}$. The modeled values of $NO_3^-$ were underestimated with NMBs of -29%. The modeled values of SOA were largely underestimated, although the correlation between modeled and measured values was 0.53. Underestimation of SOA is a common weakness of the model simulation because a fraction of SOA precursors are not included, and some key formation pathways of SOA may still be missing in current air quality models (Baek et al., 2011; Knote et al., 2014). The modeled values of SOA can thus be considered as semiquantitative evidences to facilitate analysis. Detailed evaluation results of this study are provided in the Supporting Information.

## 3. Results

### 3.1 Overview of NPF events in three campaigns

A total of 46 NPF events were observed during the three campaigns in Beijing, and the occurrence frequencies of NPF events decreased clearly in the rainy season (Table 1, Fig. 2). In Campaign 1, seven NPF events were observed during December 10–23, 2011 (7 out of 14) at the street site. During December 16–23, three NPF events were observed (3 out of 8) at the rooftop site, which occurred simultaneously with the events at the street site. In Campaign 2, seven NPF events were observed at the rooftop site during April 12–27, 2012 (7 out of 16). During April 18–27, 2012, two NPF events were observed at the street site simultaneously with the events observed (2 out of 10) at the rooftop site. In Campaign 3, 13 NPF events occurred in June out of a total of 30 observational days. The occurrence frequency decreased to approximately 20% in July and August (seven NPF events from 31 observational days). Beijing enters the rainy season in July and August, and the weather conditions are unfavorable for NPF events (Wu et al., 2007).

The NMINP varied largely from event to event in the five months, but the monthly averages were generally closer to each other. For example, the monthly average values were $1.5\pm0.8\times10^4$ cm$^{-3}$ (average value ± standard deviation) in June 2014 and $1.6\pm0.8\times10^4$ cm$^{-3}$ in July and August 2014. The large values of NMINP implied the NPF to be an important source of ambient particles when particle number concentrations were considered. The NMINP was $1.6\pm0.7\times10^4$ cm$^{-3}$ at the rooftop site in April 2012, but it was lower ($1.3\pm0.2\times10^4$ cm$^{-3}$) at the street site. In December 2011, the NMINP was only $8.3\pm4.2\times10^3$ cm$^{-3}$ at the

rooftop site, but it was still $1.5\pm0.7\times10^4$ cm$^{-3}$ at the street site. There was no significant difference in the NMINP at the rooftop site between April and June–August, but the values in the four months were significantly larger than those at the rooftop site in December (P<0.05). Zhu et al. (2017) discussed the seasonal difference in the NMINP between the two nearby sites in terms of the effects of the condensation sink and low temperature.

## 3.2 Season-dependent growth patterns of newly formed particles

As reported by Dusek et al. (2006), atmospheric particles with a diameter below 60 nm were unlikely to be activated as CCN at a normal SS, such as ≤0.2%. Investigating the growth behaviors of newly formed particles, three growth patterns, Classes I, II, and III, were identified on the basis of their potentials to act as CCN at normal SS (Figs. 3–5 and Figs. S3–6). Class I was characterized by no apparent particle growth. For example, the fitted $D_{pg}$ of new particles was almost constant at 11 nm for ~10 h on April 25, 2012, until the new particle signal dropped to a negligible level (Table 1, Fig. S3a). The new particles unlikely grew to CCN size prior to removal from ambient air.

Class II was characterized by the fitted $D_{pg}$ of new particles growing from $10\pm2$ nm to 20–50 nm, as shown in Fig. S3c, e, g, i. Class II can be further subclassified into four scenarios. In Scenario 1, the new particle growth lasted for a few hours with $D_{pg}$ increasing to 27–48 nm, after which it stopped (Fig. S3c). The increased size was maintained for a few hours until the new particle signal dropped to a negligible level. The growth of newly formed particles seemingly encountered a ceiling in Scenario 1, in which new particles grown at the maximum $D_{pg}$ unlikely contributed to CCN at normal SS. The ceiling prevented newly formed particles from growing to the CCN size in Scenario 1. The possibility of new particles to grow to CCN size in Scenarios 2–4 remains unknown. For example, new particles grew with $D_{pg}$ approaching 32–45 nm in Scenario 2. Afterwards, the signal of the new particles was apparently replaced by another signal of the new particles with an obviously smaller diameter (Fig. S3e). In Scenario 3, new particles grew with $D_{pg}$ increasing to 20–50 nm, and the new particle signal was overwhelmed by aged plumes. In the half or one hour switching from new particle signals to aged plume signals, $D_{pg}$ rapidly increased by dozens of nanometers (Fig. S3g), similar to those reported by Man et al. (2015). Scenario 3 was quite common in the presence of air pollutant plumes (Levy et al., 2013; Zhang et al., 2015). In Scenario 4, the $D_{pg}$ of new particles increased to 31–50 nm. Afterwards, no data were available to judge any further particle growth (Fig. S3i).

In Class III, the $D_{pg}$ of new particles experienced either a continuous increase or a noncontinuous increase with the final $D_{pgmax}$ closer to 75–120 nm (Figs. 3–5 and Figs. S4–6). Class III can be further classified into three growth patterns, which will be detailed in later sections. In addition to particle size, various factors such as chemical composition, particle mixing states, and meteorological conditions may also largely affect CCN activation of aerosols with $D_{pg}$ beyond 70 nm (Ma et al., 2016; Rose et al., 2017; Lee et al., 2019). Although new particles in Class III can grow to the CCN size, the CCN activation of grown new particles has been reported to vary case by case (Wiedensohler et al., 2009; Yue et al., 2011; Li et al., 2015; Ma et al., 2016). Overall, the $D_{pgmax}$ of grown new particles increased from Class I to Class III.

In December, all of these observed NPF events (three NPF events at the rooftop site plus seven NPF events at the street site) were subject to Class I (Table 1, Fig. 2a). In April, 3 out of 7 NPF events at the rooftop site and three NPF events simultaneously observed at the street site were subject to Class I. The remaining 4 NPF events at the rooftop site in April were subject to Class II, of which half belonged to Scenarios 3 and 4 (Table 1, Fig. 2b). In June–August, a total of 2, 14, and 11 out of 27 NPF events were subject to Class I, Class II, and Class III (Table 1, Fig. 2c–e), respectively. Newly formed particles in the summer appeared to have a high probability of growing to the CCN size, at least in 11 out of 27 NPF events. On the other hand, newly formed particles in the winter were unlikely to act as a potential source of CCN because their $D_{pgmax}$ reached only 11±1 nm. The occurrence probability of Class I events largely decreased in April and summer, with three Class I NPF events in April, no Class I NPF event in July, and one Class I NPF event each in June and August (Table 1, Fig. 2c-e). Additionally, the lack of Class III NPF events in April implied that newly formed particles were less likely to grow to the CCN size at normal SS.

Overall, the growth patterns of newly formed particles are strongly season-dependent, with a generally larger $D_{pgmax}$ in June–August. The mechanisms for the growth of newly formed particles to the CCN size in Class III are thus critical for understanding the importance of grown new particles acting as a potential source of CCN at normal SS. The new particle growth behaviors in Class III NPF events were thus analyzed in further detail, and SPs of grown particles with $D_{pgmax}$ at 50 nm and 70 nm were also estimated.

**3.3 Growth patterns of newly formed particles reaching CCN size**

Analyzing the observational results for June, July, and August 2014 separately, the occurrence frequencies of Class III NPF events in the three months were found to be very close to each other, with 4, 3, and 4 in June, July, and August, respectively. The 11 NPF events can be further classified into three particle growth patterns: one-stage particle growth, two-stage particle growth-A, and two-stage particle growth-B.

The one-stage particle growth pattern occurred in 4 out of 11 NPF events, in which the fitted $D_{pg}$ of newly formed particles continuously increased from 11 nm to 80–100 nm in 6–17 h on June 18, July 12–13, and August 25, 2014 (Fig. 3, Fig. S4). The growth of new particles stopped at ~24:00 in 3 out of 4 NPF events, while it stopped as early as ~16:00 in the last event.

Of the 11 NPF events, 4 events exhibited the two-stage particle growth-A pattern. The initial $D_{pg}$ of newly formed particles varied from 9 nm to 22 nm in different events, in which the particles grew to a larger size in the daytime, then oscillated for several hours, and eventually restarted the increase at night (Fig. 4, Fig. S5). In 2 out of 4 events, the increase in the $D_{pg}$ of newly formed particles stopped for 2–3 h in the middle period and then resumed to reach up to 75 nm at 22:00. In the other 2 out of 4 events, $D_{pg}$ remained unchanged for ~4 h in the middle period and then increased, reaching up to 110–115 nm at 1:00 the next day.

A total of 3 out of 11 NPF events exhibited the two-stage particle growth-B pattern, in which the $D_{pg}$ of newly formed

particles increased from 10–19 nm to 36–79 nm, then decreased to 24–50 nm in the next 2–4 h, and $D_{pg}$ increased again, reaching up to 84–120 nm (Fig. 5, Fig. S6). In two events, the decrease in newly formed particles occurred at approximately 18:00, for example, $D_{pg}$ from 78 nm to 52 nm at 18:00–21:22 on June 23 and $D_{pg}$ from 57 nm to 35 nm at 17:50–20:30 on July 26. However, the shrinkage occurred as early as 15:20–17:20 on June 11 with $D_{pg}$ from 38 nm to 24 nm.

## 4 Discussion

### 4.1 One-stage new particle growth to CCN size

Among the four one-stage growth NPF events, newly formed particles took the shortest time to reach the maximum size on June 18, 2014 (Fig. 3a). The NPF event was first observed at 09:20, lasting for 11 h. From 09:20 to 10:36, no apparent growth in newly formed particles was observed. The concentrations of gaseous precursors during that period may have been too low to cause a detectable growth in new particles with diameters >10 nm, similar to the observations reported in Hong Kong by Man et al. (2015). After 10:36, the $D_{pg}$ of newly formed particles increased from 14 nm to 88 nm at 15:54 with a particle growth rate of 14 nm h$^{-1}$. The ambient relative humidity (RH) was approximately 40% with an ambient temperature of approximately 30 °C (Fig. 3d), implying dry and hot conditions during the particle growth period. The observed mixing ratio of $O_x$ ($NO_2+O_3$) largely increased from ~60 ppb to ~130 ppb during the growth period, supporting the photochemical formation of secondary species to drive particle growth.

As mentioned above, the growth of newly formed particles is mainly attributed to sulfuric acid, ammonium nitrate, and secondary organic compounds (Wiedensohler et al., 2009; Riipinen et al., 2011; Zhang et al., 2012; Ehn et al., 2014; Man et al., 2015; Wang et al., 2015; Burkart et al., 2017; Lee et al., 2019; Wang et al., 2020). We therefore explore their respective contributions as follows. First, we calculated the contribution of sulfuric acid to the growth based on the observed mixing ratio of $SO_2$ and Equations 2–4. Second, we examined whether $NH_4NO_3$ freshly formed in $PM_{1.0}$ or $PM_{2.5}$ during the particle growth period. In case of no $NH_4NO_3$ formation, its contribution would not be expected. This is because an even higher product of $HNO_{3gas}*NH_{3gas}$ is required to overcome the kelvin effect and form $NH_4NO_3$ in nucleation mode and Aitken mode particles. Thus, the growth unexplained by sulfuric acid should be mainly contributed by SOA. Third, in case of $NH_4NO_3$ formation, we compared the net increase in $NH_4NO_3$ with that in SOA. It is noteworthy that this approach is limited by the uncertainty in explaining the growth because the ratios of increased $NH_4NO_3$ over increased SOA in $PM_{1.0}$ or $PM_{2.5}$ may not be the same as the ratios in nucleation mode and Aitken mode particles. In this case, the required mass of $NH_4NO_3$ or SOA to the growth was also estimated and compared with their respective net increases to facilitate the analysis.

Based on the observed mixing ratio of $SO_2$ shown in Fig. 3b and Equations 2–4, sulfuric acid was estimated to contribute < 2% to particle growth during the whole NPF period. Almost constant concentrations of $NO_3^-$ and $NH_4^+$ were observed at 11:00–14:00, implying that fresh $NH_4NO_3$ formation did not occur before 14:00 (Fig. 3c). Therefore, SOA was the dominant contributor to particle growth before 14:00, as supported by the decrease in the hygroscopicity parameter of 50 nm atmospheric

particles from ~ 0.3 to ~ 0.1 during the same event, independently reported by Wu et al. (2016). From 14:00 to 16:00, the concentrations of $NO_3^-$ and $NH_4^+$ significantly increased, accompanied by an increase in OOA by 11 $\mu g\ m^{-3}$. Assuming an increase in $NO_3^-$ because of the formation of $NH_4NO_3$, the net increase in $NH_4NO_3$ was 10 $\mu g\ m^{-3}$. Thus, the formation of $NH_4NO_3$ may also play an important role in the growth of new particles after 14:00. Zhu et al. (2014) and Man et al. (2015)

reported that $NH_4NO_3$ can be an important contributor to the growth of new particles (from 40–50 to nm to a larger size at night). Supposing that the particle growth during the entire growth period from 11:00 to 16:00 was completely driven by SOA, the required amount was estimated as 8.9 $\mu g\ m^{-3}$. The observed concentration of OOA in $PM_{1.0}$ increased by 15.5 $\mu g\ m^{-3}$ during the growth period, which could reasonably satisfy the required amount. Note that only secondary organic compounds of low volatility can support the growth of small particles, and those of high volatility may also contribute to the growth of large

particles (Ehn et al., 2014; Burkart et al., 2017). The growth of new particles stopped after 15:54 until the new particle signal gradually disappeared at ~20:00. The observed concentrations of OOA and $NO_3^-$ did not increase during the four hours, although they largely oscillated.

Another example of one-stage growth occurred on August 25, 2014, and newly formed particles took the longest time to reach $D_{pgmax}$ (Fig. 3e). RH was lower than 50%, and the ambient air temperature varied from 24°C to 31°C during the growth

period (Fig. 3h), also indicating dry and hot conditions during the particle growth period. The NPF event was observed from 07:50 on August 25, 2014 to 08:00 the next day. The new particle signal was unstable in the initial three hours because of the spatial heterogeneity of NPF.

The $D_{pg}$ of newly formed particles started to increase from 12 nm at 10:48 to 80 nm at 24:00 with a particle growth rate of 5.1 $nm\ h^{-1}$. During the period of 11:00–19:00, sulfuric acid contributed to only 6% of the increase in $D_{pg}$ from 12 nm to 51

20    nm on the basis of the observed mixing ratios of $SO_2$. Because of the lack of photochemical reactions, sulfuric acid concentrations should have been much lower during nighttime than during daytime (Petäjä et al., 2009; Lu et al., 2019).

No measured concentrations of particulate chemical species were available on that day. Their modeled concentrations in $PM_{2.5}$ were alternatively used to argue possible contributors to the growth of newly formed particles, although the uncertainty may be even larger than the use of measured particulate species in $PM_{1.0}$. The modeled concentrations of $NH_4^+$ and $NO_3^-$ were

almost constant at 11:00–18:00 (Fig. 3g), suggesting that $NH_4NO_3$ was did not freshly form to drive particle growth. Thus, SOA likely acted as the dominant contributor to particle growth.

The modeled net increase in particulate $NH_4NO_3$ was 3.6 $\mu g\ m^{-3}$ from 18:00 to 22:00, with the $D_{pg}$ of newly formed particles increasing from 47 nm to 70 nm (Fig. 3g). Assuming that the new particle growth from 18:00 to 22:00 was completely driven by $NH_4NO_3$, the required amount was estimated to be 3.1 $\mu g\ m^{-3}$. Thus, $NH_4NO_3$ may have contributed significantly to

particle growth in this period. Although the modeled concentrations of SOA increased (Fig. 3g), the uncertainty of the modeled values cannot allow quantifying its contribution. The $D_{pg}$ of newly formed particles increased from 70 nm to ~80 nm from

22:00 to 24:00 when the modeled concentrations of all species decreased because of the dilution effect. Afterwards, the new particles stopped growing until their signal gradually disappeared at 08:00 on the next day. The modeled concentrations of $NH_4NO_3$ and SOA were almost constant after 1:00 the next day, consistent with the lack of apparent growth in these large new particles.

During the two NPF events on July 12 and 13, sulfuric acid vapor was estimated as a minor contributor to particle growth (Fig. S3). The modeled results suggested that $NH_4NO_3$ were important contributors to particle growth only at nighttime (Fig. S4c, Fig. S4g). This left SOA as the major contributor to particle growth in daytime, but its contribution at nighttime was still unclear. Nevertheless, the concentrations of chemical species in nanometer particles of various sizes are required to confirm this.

**4.2 Two-stage new particle growth-A to CCN size**

       Fig. 4 and Fig. S5 show that the final $D_{pgmax}$ values of newly formed particles were 75 nm, 115 nm, 75 nm, and 110 nm on June 27, and August 6, 12 and 15, 2014, respectively. On June 27, 2014 (Fig. 4a–d), the NPF events were first observed at 09:00 and lasted for 18 h, with RH generally lower than 40%. Apparent growth of newly formed particles could not be observed from 09:00 to 10:30. The $D_{pg}$ of newly formed particles increased from ~10 nm at 10:30 to 35 nm at 15:20, with a GR of 5.2

15     nm h$^{-1}$. Using the observed mixing ratio of $SO_2$, sulfuric acid vapor was estimated to contribute to the first-stage particle growth by 3% (Fig. 4b). The constant concentrations of $NO_3^-$ observed during this period implied that $NH_4NO_3$ did not freshly form (Fig. 4c). Again, particle growth during the period, which could not be explained by sulfuric acid, should be completely driven by SOA. The required amount of SOA was estimated to be as low as 0.56 µg m$^{-3}$. The observed OOA fluctuated at 5–6 µg m$^{-3}$ during that period (Fig. 4c).

After 15:20, the $D_{pg}$ of newly formed particles stopped growing and fluctuated at approximately 35 nm for approximately two hours. The first-stage particle growth apparently encountered an upper limit. Compared with the concentrations observed before and after the two-hour period, the significantly decreased number concentrations of newly formed particles imply spatial heterogeneity of NPF on that day. In other words, much weaker atmospheric nucleation generated new particles in the upwind atmosphere within a certain spatial range, and the grown new particles at a lower number concentration were transported and

observed at the rooftop site at 15:20–17:40. The slightly decreased mixing ratios of $O_x$ during this time, which were unexpected considering a sharp increase in the observed $O_x$ after the period, imply reduced photochemical reaction activities in the upwind atmosphere at certain spatial ranges. The photochemical reaction activities during this period may be too weak to generate sufficient amounts of secondary organic and inorganic precursors to support the growth of new particles >35 nm to a larger size, and thus the growth encountered the upper limit, as shown in the diagram in the graphical abstract.

After 17:40, the $D_{pg}$ of newly formed particles started to increase from 32 nm to 75 nm at 22:30, with a GR of 9.7 nm h$^{-1}$, which nearly doubled the growth rate observed during the first growth stage. The observed mixing ratio of $O_x$ increased from

66 ppb at 17:20 to ~90 ppb at 21:20, supporting the secondary formation of chemical species to drive particle growth (Fig. 4b). The observed concentrations of OOA (left axis) and $NO_3^-$ (right axis) rapidly increased from 18:00 to 22:20, with the former being approximately four times larger than the latter. The required amount of $NH_4NO_3$ for particle growth during the period was estimated to be 5.3 µg m$^{-3}$, while the net increase in $NH_4NO_3$ was 1.6 µg m$^{-3}$. SOA may dominate the growth of new particles. Lee et al. (2016) and Huang et al. (2019) recently reported that highly functionalized organonitrates generated from the reaction of $NO_3$ free radicals with organics can contribute to the growth of particles at nighttime. After 22:30, the new particles stopped growing until their signal gradually disappeared at 03:00 on the next day.

Following the analysis mentioned above, freshly formed SOA was argued to dominantly drive the first-stage particle growth on August 6 (Fig. 4e), 12, and 15, 2014 (Fig. S5). On the other hand, newly formed $NH_4NO_3$ likely acted as an important contributor to second-stage particle growth. Again, large uncertainties in modeled concentrations may exist because of the lack of direct measurements of chemical species in nanometer particles of various sizes.

### 4.3 Two-stage new particle growth-B to CCN size

Among the three two-stage growth-B NPF events, the longest shrinkage (approximately 4 h) in grown new particles occurred on June 23, 2014 (Fig. 5a). According to our analysis, the first-stage particle growth on that day was driven by SOA because the estimated sulfuric acid and observed $NO_3^-$ plus $NH_4^+$ yielded either a small percentage or negligible contribution to particle growth. The $D_{pg}$ of newly formed particles increased from 17 nm at 11:20 to 79 nm at 17:20, with a GR of 10 nm h$^{-1}$. From 11:20 to 17:20, the mixing ratio of $O_x$ increased from 74 ppb to 122 ppb (Fig. 5b). The net increase in the observed OOA was 12.2 µg m$^{-3}$ during this period (Fig. 5c), while the required amount of SOA was estimated as 4.1 µg m$^{-3}$. SOA was very likely to be the major contributor to particle growth in this period. As independently reported by Wu et al. (2016), the hygroscopicity parameter of 50 nm atmospheric particles decreased from ~0.15 to ~0.05 during the same event.

The $D_{pg}$ of newly formed particles stopped growing at 79 nm from 17:20 through 18:00 and then decreased from 79 nm to 52 nm at 21:22, with a decrease rate of 8 nm h$^{-1}$. During this period of shrinkage, the observed mixing ratio of $O_x$ largely decreased from 130 ppb to 80 ppb, and the observed OOA decreased from 16.2 µg m$^{-3}$ to 11.4 µg m$^{-3}$ (Fig. 5b–c). However, the concentrations of $NH_4^+$ were almost constant. Repartition of the semivolatile SOA in gas and particle phases was hypothesized to cause the evaporation of semivolatile SOA to the gas phase. The shrinkage may also be argued as being attributable to the spatial heterogeneity of NPF, but modeling of size-segregated number concentration is required to confirm this.

After 21:22, $D_{pg}$ restarted to increase from ~50 nm to 90 nm over 4 h. The formation of $NH_4NO_3$ likely yielded an important contribution to the second stage of particle growth—a net observed increase of 4.5 µg m$^{-3}$ versus the required amount of 8.4 µg m$^{-3}$. SOA may also contribute to the second stage of particle growth according to a net increase in OOA by 1.5 µg m$^{-3}$ (Fig. 5c). After the second stage of growth, the $D_{pg}$ of new particles experienced small oscillations at ~90 nm until the

signal was overwhelmed completely by aged plumes.

Following similar observations on June 23, reduced photochemical reaction activities were also argued to cause the shrinkage in newly formed particles on June 11 and July 26, 2014 (Fig. S6). The observed and modeled results for the two days imply that $NH_4NO_3$ played an important role in new particle growth only at night. In the daytime, SOA likely acted as the major contributor.

## 4.4 Statistical analysis of factors related to new particle growth

The growth rate of newly formed particles is an intensity quantity and mainly determined by the concentrations of condensable vapors such as sulfuric acid, organics of various volatilities, nitric acid, and ammonia (Zhang et al., 2012; Ehn et al., 2014; Man et al., 2015; Lee et al., 2019). In contrast, $D_{pgmax}$ values are determined by the total amount of vapors condensed on grown new particles (an extensive quantity), which may or may not have a positive correlation with the concentrations of these vapors (Zhu et al., 2019). The values of $D_{pgmax}$ were plotted against those of GR in Fig. 6a (two variables during the first growth period were used to determine the occurrence of two-stage particle growth) and found to be widely scattered with r=0.23. When three circled points were excluded, $D_{pgmax}$ was significantly correlated with GR, but the r value was still as low as 0.48 (Fig. 6a). GR alone is not sufficient to characterize the growth of newly formed particles considering their potential impacts on the climate, and both $D_{pgmax}$ and GR should be alternatively used.

As mentioned above, SOA and $NH_4NO_3$ are likely two major contributors to particle growth in different periods of NPF events, with small contributions of sulfuric acid. Fig. 6b shows the net hourly increases in OOA and $NH_4NO_3$ against the hourly required masses for particle growth, assuming densities of 1.5 $\mu g$ $m^{-3}$ for OOA and 1.7 $\mu g$ $m^{-3}$ for $NH_4NO_3$. Both OOA and $NH_4NO_3$ generally increase with increasing required masses and reasonably satisfy the required masses, but they are largely scattered in Fig. 6b. It remains challenging to accurately quantify the contributors to the growth of newly formed particles.

The generation of OOA and $HNO_3$ is strongly related to oxidation reactions during the daytime. Thus, we further plotted $D_{pgmax}$ and GR against $O_x$ ($O_x=NO_2+O_3$) in the particle growth period during the daytime. Fig. 6c shows a good correlation between $D_{pgmax}$ and $O_x$ (hourly average value when $D_{pgmax}$ reached) with r=0.80 and P<0.01. The values of $O_x$ in Class I NPF events were significantly smaller than those in Class II and Class III with P<0.05, and the lower $O_x$ could be one of the factors for the lack of apparent particle growth in Class I. In addition, there was no significant difference in $O_x$ between Class II and Class III. Including $O_x$, other factors, such as condensational sink, should also affect the particle growth in Class I, II, and III NPF events (Guo et al., 2020). Fig. 6d shows a significant correlation between GR and $\overline{O_x}$ (average value during the entire growth period) with r=0.67 and P<0.01. The decreased r value implies that the response of GR to the increase in $O_x$ is highly variable.

Oxidation products of biogenic VOCs, such as highly oxygenated molecules (HOMs), have been reportedly overwhelmed

to determine the condensation growth of newly formed particles in the small size range because of their low volatilities (Ehn et al., 2014; Lee et al., 2019). In this study, the clear seasonal boundary of Class I and Class II + III NPF events—for example, 100% of Class I events in winter versus 7% and 93% of Class I and Class II+III events in summer—also points toward the importance of oxidation products of biogenic VOCs in particles growing from ~10 nm to larger sizes. In the summertime, theoretically increased emissions of biogenic VOCs and enhanced photochemical reactions indicated by $O_x$ are expected to generate more HOMs for the growth of particles from ~10 nm to larger sizes. In spring, approximately half of the NPF events are subject to Class I. However, there were no Class III events. The distinctive seasonal particle growth patterns may further imply that the amount of oxidation products of biogenic VOCs not only determines the growth of new particles from ~10 nm to larger sizes, but also the CCN size. However, direct measurements of HOMs in small-sized nanoparticles were unavailable to support this argument. In fact, such measurements remain challenging among the research community, as reviewed by Lee et al. (2019).

### 4.5 SP during Class III NPF events

The potential contribution of new particles to the population of CCN was evaluated using the calculated SP. For Class III NPF events, the final SPs are listed in Table 1. For example, in the one-stage growth NPF event on August 25, 2014 (Fig. S2a–b), the $N_{50-200\ nm}$ gradually decreased from ~05:00 to ~11:00 and reached the lowest values of $787\pm120$ cm$^{-3}$ at 10:18-11:59. The $N_{50-200\ nm}$ then increased with the growth of newly formed particles. It reached the highest values of $9210\pm257$ cm$^{-3}$ at 19:47-20:17 if a few spikes afterwards were not considered. The spikes were more likely due to the intrusion of plumes rather than the new particle signal. The lowest values over half an hour were considered as the regional background. The mean of the highest values over half an hour minus the mean of the lowest values was considered as the net maximum number concentration of grown new particles beyond 50 nm, which was $8.4\times10^3$ cm$^{-3}$. Thus, the $SP_{50\ nm}$ was estimated to be 35%. Using the similar approach as shown in Fig. S2b, the net maximum number concentration of grown new particles beyond 70 nm was $4.7\times10^3$ cm$^{-3}$, and the estimated $SP_{70\ nm}$ was 20%. In the two-stage new particle growth on June 27, 2014 (Fig. 4a), the net maximum number concentration of grown new particles beyond 50 nm reached a maximum value of $9.0\times10^3$ cm$^{-3}$, with the estimated $SP_{50\ nm}$ to be 60%. Using the same method, the $SP_{70\ nm}$ was estimated to be 39%.

Overall, in Class III NPF events, the $SP_{50}$ varied from 35% to 60%, with a median of 50%. Meanwhile, the $SP_{70}$ varied from 20% to 44%, with a median of 32%. Our results imply that an appreciable fraction of new particles can grow to CCN size prior to being removed by atmospheric processes. Considering that high SS occasionally occurs in the atmosphere (Fan et al., 2018), new particles with $D_{pg}$ increasing up to 50 nm may also be activated as CCN. Thus, > 30% of new particles in 11 out of 27 NPF events in the summer of 2014 can reach 50 nm and may eventually contribute to the population of CCN.

**4.6 Spatial heterogeneity of NPF**

The spatial heterogeneity of NPF can be clearly identified using high time-resolution measurements. Two NPF events were used as examples to demonstrate the spatial heterogeneity.

The NPF event on August 6, 2014 (Fig. 4e) clearly exhibited spatial heterogeneity; the signal of new particles largely dropped to a negligible level approximately one hour after 11:37 and then increased to a detectable level (Fig. 4e). At approximately 17:40, $D_{pg}$ jumped from 25 nm to 50 nm within five min, indicating a large spatial heterogeneity before and after 17:40–17:50. New particles observed after 17:51 were hypothesized to experience a growth similar to the trend in the white dashed line (Fig. 4e) in the upwind atmosphere during the period from 11:37 to 17:51.

Moreover, both the number concentrations and $D_{pg}$ of new particles exhibited an inverted bell-shape at 23:00–01:51 on August 6, 2014 (Fig. 4e). The inverted bell-shape very likely reflects the spatial heterogeneity of NPF in the upwind atmosphere at a certain spatial range. The new particle signal was clearly enhanced after 01:51 on August 7, 2014. The new particles observed after the time were hypothesized to experience a growth similar to the trend in the white dashed line (Fig. 4e) in the upwind atmosphere.

The NPF event on June 23 also exhibited clear spatial heterogeneity (Fig. 5a). From 12:00 to 18:00, $N_{8\text{-}200nm}$ oscillated at $1.2\pm0.2\times10^4$ cm$^{-3}$ (Fig. 5b). In approximately 20 min, $N_{8\text{-}200nm}$ increased to a higher level and then oscillated at $1.5\pm0.2\times10^4$ cm$^{-3}$ from 18:20 on June 23 to 01:30 on June 24. $N_{8\text{-}200nm}$ then oscillated at $1.0\pm0.1\times10^4$ cm$^{-3}$ from 01:50 to 04:15 on June 24.

Based on the time series of new particle number concentrations and their sizes observed, the spatial heterogeneity of NPF can be inferred to have occurred universally in each NPF event. This phenomenon should be considered for accurately evaluating the climate impacts of NPF events.

**5 Conclusions**

In this study, we investigated 46 NPF events in Beijing's urban atmosphere through three campaigns, with particular attention to the growth behaviors of newly formed particles. First, we found seasonal variations in the maximum sizes of newly grown particles. For instance, $D_{pgmax}$ was found to exceed 75 nm in 11 out of 27 NPF events in summer. However, no apparent growth in new particles with $D_{pgmax} < 20$ nm was observed in December across all events, which could be attributed to several factors, such as the lower level of $O_x$ and high condensation sink. Correlation analyses also suggest that the concentrations of $O_x$ may play an important role in determining $D_{pgmax}$. This finding may allow us to rethink the seasonal impacts of NPF events on the climate in Beijing and other urban areas in northern China.

According to the observed mixing ratio of SO$_2$, sulfuric acid vapor generally yielded minor contributions to the growth of new particles. The observed and modeled concentrations of particulate chemical species suggested that the growth of newly formed particles during the daytime was mainly caused by OOA (or SOA). At night and late afternoon, the increased amount of NH$_4$NO$_3$ can reasonably support new particle growth in most Class III NPF events. Secondary organics were also an

important contributor to nighttime new particle growth in Class III NPF events with OOA to be observed, but its contribution was still unclear in cases absent of OOA observations. Nevertheless, direct measurements of secondary organics in nanometer particles of different sizes are required to confirm their contribution.

To verify the climate impacts of NPF events, the $SP_{50}$ and final $SP_{70}$ need to be quantified. In Class III NPF events, the $SP_{50}$ and final $SP_{70}$ varied from 35% to 60% and from 20% to 44%, respectively, implying that an appreciable fraction of new particles can grow to CCN size. Our observations also indicated that each NPF event exhibited spatial heterogeneity to some extent, which may be attributable to varying photochemical reaction activities. When photochemical reaction activities are low, the growth of new particles may reach an upper limit or even decline. These factors should also be considered for accurately evaluating the climate impacts of NPF events in the future.

**Data availability.** The research data can be accessed upon contact with the corresponding author (Xiaohong Yao (xhyao@ouc.edu.cn) and Yujiao Zhu (zhuyujiao@sdu.edu.cn)).

**Author contributions.** XY designed the research. YZ, MZ and YS conducted the field measurements. LH and XL run the CMAQ model. LM and YZ analyzed the data and wrote the paper. YG, YS, HG and XY helped to interpretation of the results. XY revised the original manuscript. All authors contributed toward improving the paper.

**Competing interests.** The authors declare that they have no conflict of interest.

### Acknowledgment

This research was supported by the National Key Research and Development Program in China (grant no. 2016YFC0200504), the Natural Science Foundation of China (grant no. 41576118 and grant no. 41430646).

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

**Abstract Art**

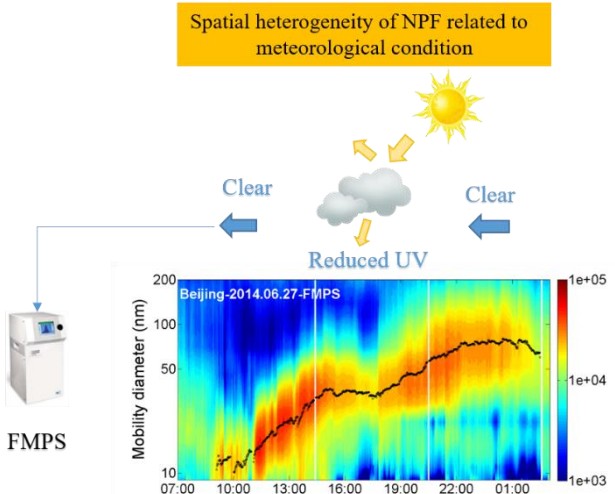

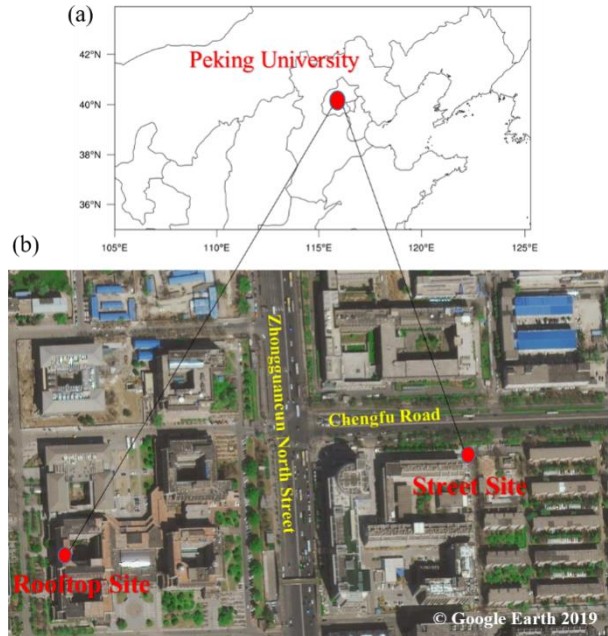

**Fig. 1** Locations of sampling sites (a) and satellite imagery of the two sampling sites (b) (downloaded from https://www.earthol.com/).

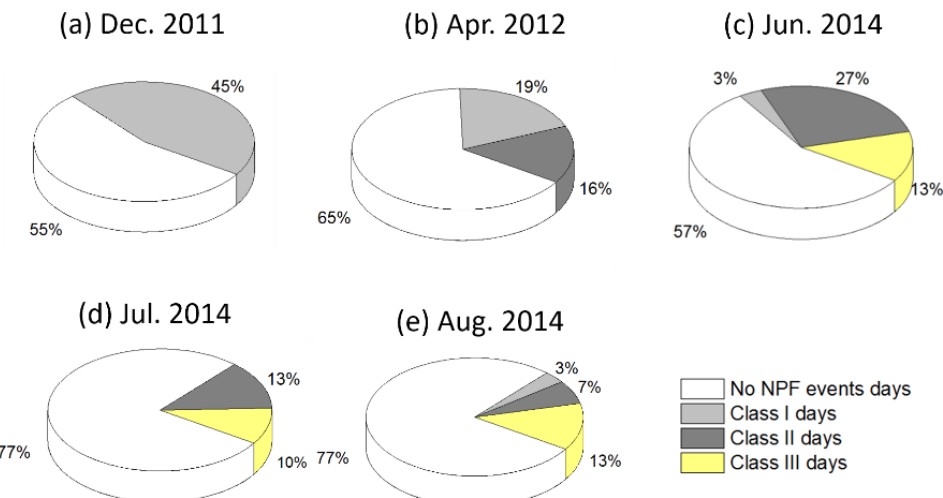

**Fig. 2** Occurrence frequencies of difference growth patterns of NPF events in observational months.

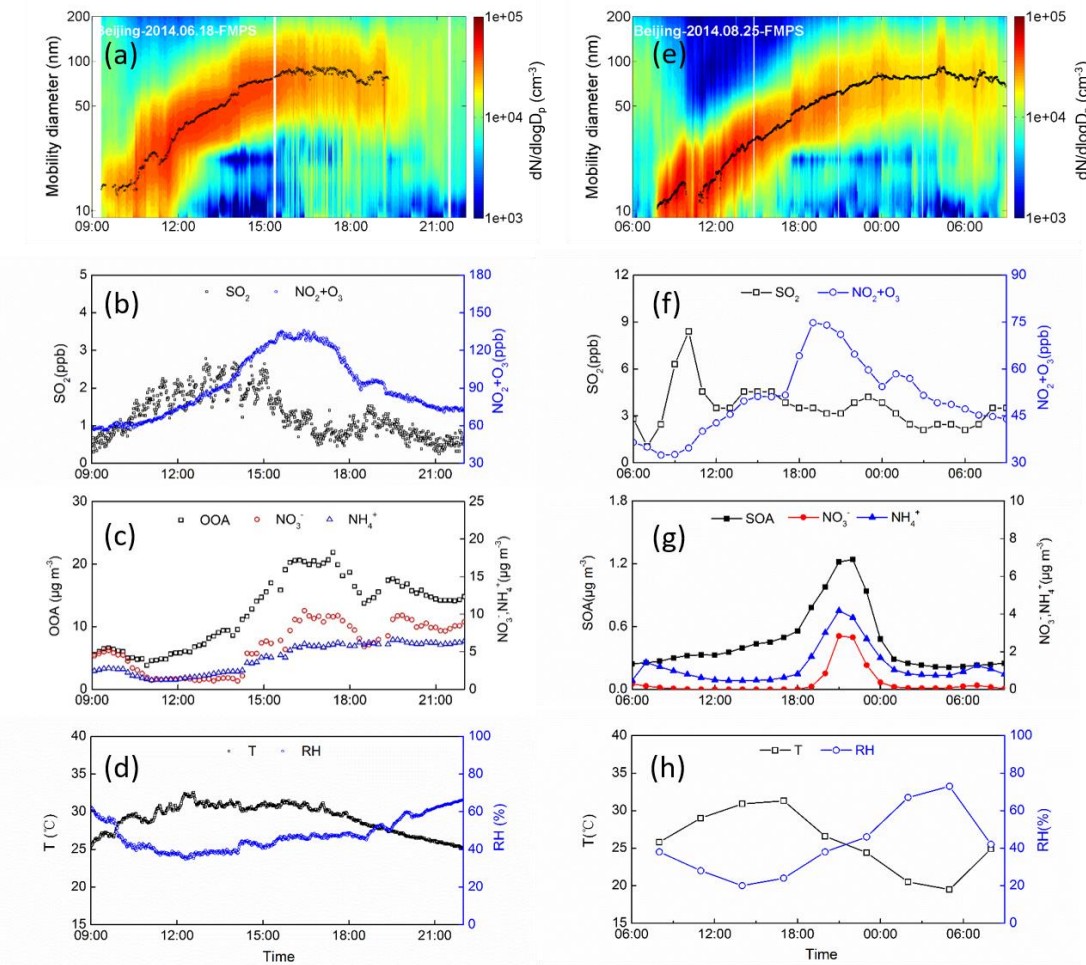

**Fig. 3** NPF events on June 18 and August 25, 2014 ((a, e) contour plot of particle number concentration; (b, f) time series of observed mixing ratios of $SO_2$ and $NO_2+O_3$; (c) time series of observed OOA, $NO_3^-$ and $NH_4^+$ in $PM_{1.0}$; (d, h) time series of ambient T and RH; (g) time series of modeled SOA, $NO_3^-$ and $NH_4^+$ in $PM_{2.5}$).

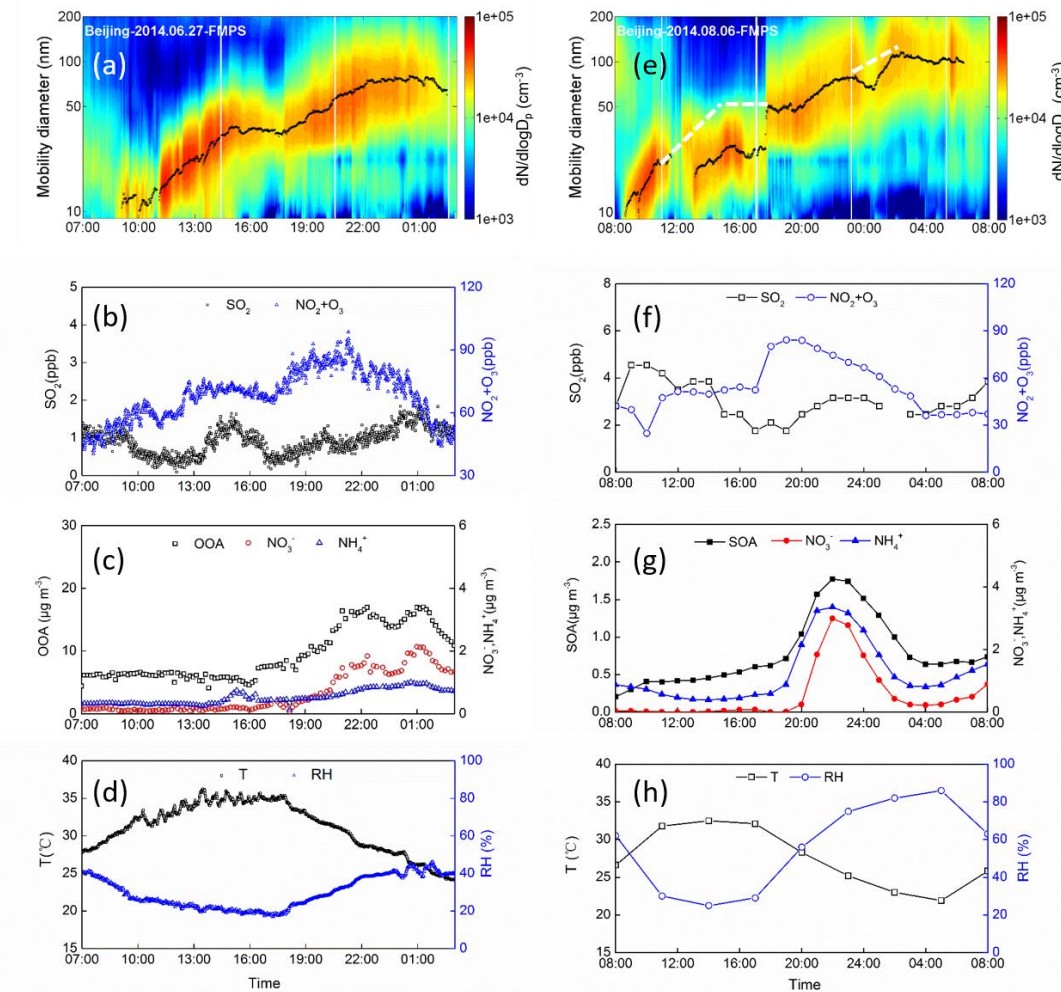

**Fig. 4** NPF events on June 27 and August 6, 2014 ((a, e) contour plot of particle number concentration; (b, f) time series of observed mixing ratios of $SO_2$ and $NO_2+O_3$; (c) time series of observed OOA, $NO_3^-$ and $NH_4^+$ in $PM_{1.0}$; (d, h) time series of ambient T and RH (g) time series of modeled SOA, $NO_3^-$ and $NH_4^+$ in $PM_{2.5}$).

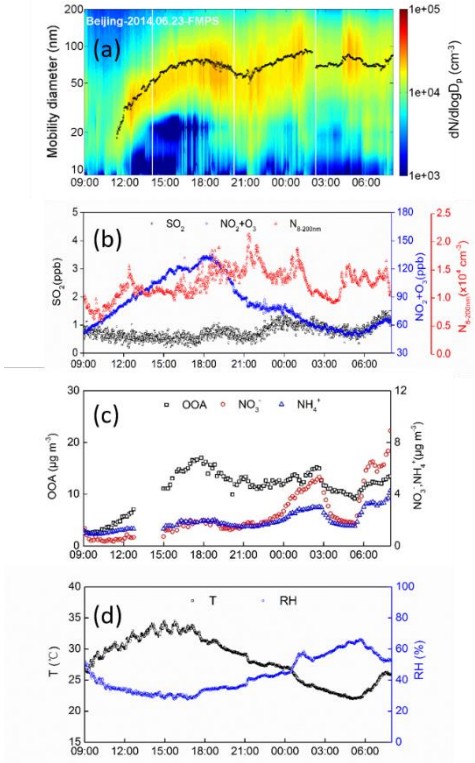

**Fig. 5** NPF event on June 23, 2014 ((a) contour plot of particle number concentration; (b) time series of observed mixing ratios of $SO_2$, $NO_2+O_3$, and $N_{8-200nm}$; (c) time series of observed OOA, $NO_3^-$ and $NH_4^+$ in $PM_{1.0}$; (d) time series of ambient T and RH).

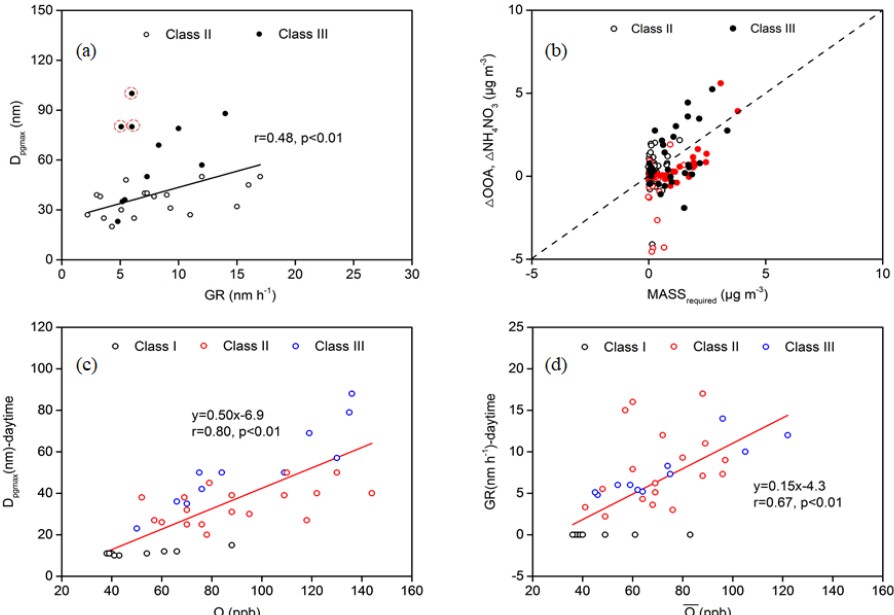

**Fig. 6** Relationship between $D_{pgmax}$ and GR (a); hourly variations in measured OOA (black mark) and $NH_4NO_3$ (red mark, assuming $NO_3^-$ to be completely associated with $NH_4^+$) versus required masses of OOA and $NH_4NO_3$ for corresponding particle growth (b); relationship between $D_{pgmax}$ in the daytime and the corresponding maximum $O_x$ (c), and relationship between GRs in the daytime and the average mixing ratio of $O_x$ (d).

**Table1 Characteristics of NPF events in Beijing**

| Season | Date | Period | GR or SR (nm h$^{-1}$) | NMINP (10$^4$#cm$^{-3}$) | $D_{pgmax}$ (nm) | SO$_2$ (ppb)[d] | SP (%) | O$_3$+NO$_2$ (ppb)[g] |
|---|---|---|---|---|---|---|---|---|
| Winter | 10 Dec 2011* | 9:00-15:00 | - | 1.5 | 11 | 2.5~4.4 | - | 36~38 |
| | 11 Dec 2011* | 11:00-14:00 | - | 2.5 | 11 | 7.2~16 | - | 34~40 |
| | 14 Dec 2011* | 10:00-16:00 | - | 1.1 | 11 | 3.1~5.8 | - | 31~39 |
| | 15 Dec 2011* | 10:30-17:30 | - | 1.0 | 11 | 1.6~5.3 | - | 33~39 |
| | 21 Dec 2011 | 13:00-18:00 | - | 0.5 | 10 | 1.4~3.5 | - | 33~43 |
| | 21 Dec 2011* | 9:00-12:00 | - | 1.1 | 12 | 2.5~5.9 | - | 22~61 |
| | 22 Dec 2011 | 10:00-15:00 | - | 1.3 | 10 | 2.3~6.0 | - | 32~41 |
| | 22 Dec 2011* | 11:30-15:40 | - | 2.3 | 10 | 2.3~7.6 | - | 23~41 |
| | 23 Dec 2011 | 11:00-14:00 | - | 0.7 | 10 | 3.5~16 | - | 32~43 |
| | 23 Dec 2011* | 9:40-16:30 | - | 0.7 | 10 | 3.6~9.2 | - | 38~41 |
| Spring | 12 Apr 2012 | 9:20-18:20 | 2.2 | 2.9 | 27 | 1.0~2.3 | - | 41~57 |
| | 13 Apr 2012 | 11:20-19:00 | 6.2 | 1.5 | 25 | 1.4~3.6 | - | 61~76 |
| | 14 Apr 2012 | 12:00-19:00 | 9.3 | 0.8 | 31 | 2.0~7.7 | - | 73~88 |
| | 15 Apr 2012 | 11:30-19:00 | - | 1.7 | 12 | 0.0~2.1 | - | 57~66 |
| | 16 Apr 2012 | 10:22-14:20 | 7.9 | 1.2 | 38 | 1.3~3.7 | - | 52~69 |
| | 25 Apr 2012 | 10:07-20:00 | - | 1.0 | 11 | 0.0~1.9 | - | 47~54 |
| | 25 Apr 2012* | 10:07-20:00 | - | 1.1 | 11 | 0.0~1.9 | - | 47~54 |
| | 27 Apr 2012 | 9:40-16:00 | - | 2.1 | 15 | - | - | - |
| | 27 Apr 2012* | 9:40-16:00 | - | 1.4 | 15 | - | - | - |
| Summer | 1 Jun 2014 | 12:00-16:00 | - | 1.1 | 15 | 0.4~1.5 | - | 77~88 |
| | 3 Jun 2014 | 8:00-12:00 | 4.3 | 3.0 | 20 | 1.2~10 | - | 56~78 |
| | 4 Jun 2014 | 11:30-22:00 | 11 | 1.2 | 27 | 1.2~3.7 | - | 67~118 |
| | 7 Jun 2014 | 9:00-(+1) 3:00 | 5.5 | 1.3 | 48 | 0.0~1.3 | - | 32~64 |
| | 8 Jun 2014 | 9:00-14:00 | 12 | 1.5 | 50 | 3.5~9.0 | - | 41~110 |
| | 9 Jun 2014 | 10:55-19:40 | 7.1 | 1.1 | 40 | 1.0~4.5 | - | 55~122 |
| | 11 Jun 2014 | 9:20-(+1) 3:20 | 5.4/5.1[a]/9.0[b] | 1.1 | 36/84[c] | 0.0~1.2 | 54/36[e] | 43~89 |
| | 12 Jun 2014 | 8:00-15:00 | 3.6 | 3.1 | 25 | 1.2~7.3 | - | 50~87 |
| | 18 Jun 2014 | 9:20-20:20 | 14 | 1.8 | 88 | 0.4~2.8 | 56/39[e] | 56~136 |
| | 23 Jun 2014 | 11:20-(+1) 1:22 | 10/8.0[a]/10[b] | 0.5 | 79/90[c] | 0.2~1.3 | N/A[f] | 53~135 |
| | 27 Jun 2014 | 9:00-(+1) 3:00 | 5.2/9.7[b] | 1.5 | 35/75[c] | 0.1~1.9 | 60/39[e] | 44~99 |
| | 28 Jun 2014 | 7:00-19:00 | 3.0 | 1.8 | 39 | 0.6~10 | - | 15~106 |
| | 29 Jun 2014 | 8:50-15:00 | 7.3 | 0.7 | 40 | 1.7~9.2 | - | 62~144 |
| | 8 Jul 2014 | 9:30-21:00 | 16 | 1.0 | 45 | 0.9~4.3 | - | 52~79 |
| | 9 Jul 2014 | 10:00-17:30 | 15 | 2.4 | 32 | 0.9~4.3 | - | 42~91 |
| | 12 Jul 2014 | 9:00-(+1) 4:00 | 6.0 | 1.6 | 80 | 1.8~3.5 | 46/27[e] | 36~84 |
| | 13 Jul 2014 | 7:30-(+1) 4:00 | 6.0 | 2.5 | 100 | 2.1~5.6 | 39/22[e] | 37~78 |
| | 14 Jul 2014 | 8:00-20:00 | 17 | 2.5 | 50 | 3.2~5.3 | - | 41~135 |
| | 25 Jul 2014 | 11:20-22:00 | 9.0 | 0.6 | 39 | 3.9~6.0 | - | 86~109 |
| | 26 Jul 2014 | 14:33-(+1) 8:00 | 12/8.2[a]/7.5[b] | 0.9 | 57/120[c] | 5.3~11 | 46/23[e] | 23~130 |
| | 6 Aug 2014 | 8:41-(+1) 8:00 | 4.8/10[b] | 1.4 | 23/115[c] | 1.4~4.6 | 54/44[e] | 25~82 |
| | 12 Aug 2014 | 10:00-22:00 | 7.3/6.5[b] | 1.3 | 50/75[c] | 2.1~4.6 | N/A[f] | 47~109 |
| | 15 Aug 2014 | 10:10-(+1) 3:42 | 8.3/8.7[b] | 0.9 | 69/110[c] | 2.1~6.3 | N/A[f] | 41~119 |

| | | | | | | | | |
|---|---|---|---|---|---|---|---|---|
| | 24 Aug 2014 | 8:00-19:00 | 3.3 | 3.0 | 38 | 3.9~8.1 | - | 30~52 |
| Summer | 25 Aug 2014 | 7:50-(+1) 9:00 | 5.1 | 2.4 | 80 | 2.1~8.4 | 35/20[e] | 32~75 |
| | 26 Aug 2014 | 9:00-23:00 | 5.1 | 1.1 | 30 | 0.7~7.0 | - | 44~95 |
| | 27 Aug 2014 | 12:25-14:30 | - | 1.1 | 12 | - | - | - |

[*]: The NPF events occurred on the street site.

[a]: Refers to the shrinkage rates of two-stage growth-B.

[b]: Refers to the second-stage growth rates.

[c]: Refers to the $D_{pgmax}$ of the second-stage growth.

[d]: Refers to the mixing ratio range of $SO_2$ during the NPF period.

[e]: Refers to the SP with the $D_{pg}$ increasing up to 50 nm and 70 nm, respectively.

[f]: Refers to Equation 5 not applicable for calculating SP during the NPF events.

[g]: Refers to the mixing ratio range of $O_x$ ($NO_2+O_3$) during the NPF period.