# Peer review of "Investigating three patterns of new particles growing to the size of cloud condensation nuclei in Beijing's urban atmosphere"

_Atmospheric Chemistry and Physics, 2019_

## Referee Comment (RC1) · Anonymous Referee #2 · 26 May 2020

This study investigated seasonal variations of new particle formation (NPF) events in Beijing by using observations of particle size distributions and chemical compositions of aerosols and numerical model simulations. The authors found no apparent growth of new particles in winter whereas the growth of new particles to CCN size (50 or 75 nm) was often observed in summer. The three patterns of NPF events during the summertime were discussed in terms of secondary aerosol formation, evaporation of semi-volatile species, and spatial heterogeneity of NPF events.

The scope of this manuscript is well suited to ACP, and the data obtained by the authors are valuable and important to understand the mechanisms of NPF events in urban

atmospheres. However, the current manuscript needs substantial revisions before the manuscript is considered as a publication of ACP as shown below.

1) Page 1, Line 17:

"11/27" should be revised. For example, "11 new particle formation (NPF) events out of 27 events" may be better. Other parts written similarly in the text should also be revised.

2) Page 2, Lines 21-28:

The authors described what they did in this study. However, it is not clear to me which parts of this manuscript are scientifically new. There are many previous studies on NPF in Beijing and other urban areas. The authors should summarize these previous studies and describe what are well understood and what are poorly understood in Introduction. Then, the objectives of this study should be described more clearly.

The sentence at Lines 18-20 (Thus far, which chemicals drive the growth..) is a point poorly understood, but I don't think the understanding on this point was improved by this study.

3) Page 4, Line 3: Equation (2)

Please add descriptions on the uncertainty of this equation.

4) Page 4, Line 11:

The SPR analysis (section 4.5) is not meaningful. It is hard to quantitatively estimate the survival fraction of new particles from this equation because the SPR values can be greater than 100% in many cases (Table 1). I think the authors may be able to calculate the loss rate of new particles during each NPF event from CS.

5) Page 4, Line 13:

Please clarify why 3 sigma was chosen.
6) Page 4, Line 20: Mass$_{required}$

The authors compared Mass$_{required}$ with the changes in mass concentrations of organic and nitrate aerosols, but the latter is generally controlled by accumulation mode particles, not nucleation mode particles. The comparison between Mass$_{required}$ (changes in aerosol mass for nucleation or Aitken mode particles) and the changes in mass concentrations of organic and nitrate aerosols (mainly controlled by accumulation mode particles) is therefore not so meaningful (in sections 4.1-4.3).

7) Page 5, Lines 1-2:

Please provide some brief descriptions on model setups.

8) Pages 5, Lines 8-9:

Please describe on model evaluations more clearly (e.g., the degree of agreement with observations, chemical species evaluated).

9) Page 5, Lines 23-27:

The unit of number concentrations in this paragraph is probably not correct.

10) Page 6, Lines 2-19:

Please clarify why Class II was subclassified to 4 scenarios. What is the purpose of this?

11) Page 8, Line 3: the contribution of <2%

Please clarify how the authors estimated this contribution.

I think the authors have sulfate data observed by AMS. The data can be shown like OM and nitrate in Figures 2-4.

12) Page 8, Line 7: 13 ug m-3

Please clarify how the authors estimated this value. Did the authors consider the

spread of particle size distributions? (like 3 sigma in equation (3)).

13) Page 8, Lines 6-8:

As I described above, the comparison between the required mass (13 ug m-3) and PM1 enhancement (15 ug m-3) is not so meaningful because the former focuses on nucleation/Aitken mode particles but the latter is usually dominated by accumulation mode particles.

I think what the authors can do here is to calculate mass concentration changes for sulfate, nitrate, ammonium, and SOA and to discuss which changes are the largest during the growth periods of new particles.

14) Page 8, Line 7:

OM can be divided into HOA (POA like) and OOA (SOA like) by using m44 and m57 signals. Only OOA can contribute to the growth of particles.

15) Page 9, Lines 3-4:

I don't agree with this authors' description. The simulated OA and nitrate cannot be used to interpret the data unless the authors evaluate the simulations with observations.

16) Page 10, Line 16

Delete "(ON)".

17) Page 10, Line 28:

OM can be divided to HOA and OOA as I described above.

18) Page 11, Lines 5-6: "Repartition of the . . ."

This part should be removed because no data can support this sentence.

19) Page 12, Line 19

"then in", "transience": they should be corrected.

20) Page 12, Line 26: Section 4.5

As I described above, this section is not so meaningful and should be removed.

How did the author consider the contribution of primary particles in this analysis?

---

## Referee Comment (RC2) · Anonymous Referee #1 · 30 May 2020

This study investigated patterns and characteristics of atmospheric new-particle formation events in Beijing. The authors categorized these observed events into three classes based on the number size distributions of the newly-formed particles. Further, by combing the size distribution with the speciation of measured or modelled gas- and particle-phase pollutants, the authors discussed the contribution of organic and inorganic compounds to particle growth during different type of events.

The manuscript focus on the topic of new-particle formation in the urban atmosphere, trying to address critical questions that whether or not the newly-formed particles can grow to the CCN size, and what conditions/species control the grow process. The

scope of the manuscript is thus suitable for ACP, and the data the authors presented are ample and interesting. However, the interpretation of some key results is questionable and leads to unrigorous conclusions. Major revisions and improvements are needed before this manuscript can be considered for publication as an ACP paper.

Major comments:

1) The authors defined three classes. "Class I was characterized by no apparent particle growth" makes sense, this class might indicate either a lack of supersaturated condensable vapors so that particles don't grow, or a too high condensation sink so that small particles don't survive. But is there a better way to classify the rest events? Particles are larger than 50 (or 75) nm doesn't necessarily mean they are good CCNs; and there are so many factors (chemical, physical or meteorological) that can determine whether or not the particles grow over 50 (or 75) nm. Classifying the events just based on the "cut-off" size doesn't really help modelers or lab experimentalists to understand the real atmosphere. Please justify the classification or improve it.

2) From the surface plot of these NPF events (e.g. Fig 2a, e; Fig 3a, e; Fig 4a), I don't see any significant band of pre-existing particles. Were these events all observed in very clean days? Or is it because the linear color scale veil the background particles? Please do change to the log color scales.

3) The author stated that many growth events lasted for over 10 hrs or even a whole day. Was there any primary emission mixing with the newly-formed particles, e.g. from vehicles, restaurants or factories? Is it true that there was only condensational growth without mixing during the whole period? Please discuss this and also show O:C from the AMS measurement to verify the statement.

4) About the AMS measurements, the sampling site is 8 km away from Peking University, how long does it take for an air parcel transport from one site to the other? Roughly one hour maybe? How well does the AMS result represent the particle composition at Peking University? I think this question need to be better addressed in order to discuss

the spatial heterogeneity.

5) AMS measured the bulk PM_1.0, how well does the chemical composition in PM_1.0 represent the species drive the sub-100 nm particle growth? Were there any aerodynamic diameters measured by AMS at the same time? Discuss more about the uncertainty here.

6) Page 4, line 20, Equation 4, I don't find the exact same equation in the references the authors cited here. Using averaged particle number concentration over the whole growth period can bring in large uncertainties. Newly-formed particle are prone to coagulational loss; this means particle number concentration at D_pg1 will be much higher than that at D_pg2, and the Mass_required will be overestimated. Please justify the equation, or calculate the particle mass concentration for each FMPS scan and also take the coagulation into account.

7) Was there a special reason to sum up O3 and NO2 instead of discussing them separately? NO2 is not always associated with O3, it could come from primary emission such as vehicle exhaust.

8) The authors briefly mentioned seasonal variation, but didn't dive into the details in, for example, wintertime events. A recent study (Wang et al., 2020, Nature 581 (7807), 184-189) show that NH4NO3 could help the newly-formed particles grow and survive in winter. So it's intriguing to know that the authors observed that the newly-formed particles didn't grow during wintertime events, but it would be more important to understand why they didn't. Was it because of a lack of supersaturated condensable vapors, or a too high condensation sink?

9) Please discuss more about the model uncertainty, sensitivity test, etc.

10) The authors used many sentences describing the particle growth/shrinkage processes vs time, e.g. "However, the shrinkage occurred as early as 15:20-17:20 on 11 June.". Yet these sentences contain very limited information. I would suggest the

authors go through the whole manuscript and reword these sentences by discussing more deeply about the environmental conditions or the causes of these different types of growth.

Minor comments:

1) Page 1, line 17: "11/27" (and hereinafter) should be something like "11 out of 27".

2) Page 1, line 22: "... in the remaining NPF events", please add the number here.

3) Page 2, line 26: "survival probability ratios", "survival probability" would be better.

4) Page 3, line 9: should the coefficient be size-dependent?

5) Page 3, line 13: "During the other observational periods...", please specify the date.

6) Page 6, line 8: "to a negligible level; in Scenario 2", the semicolon should be period.

7) Page 6, line 10: "... may not represent two NPF events occurring in one day.", why? They look very much like two events.

8) Page 6, line 14: "... associated with wind direction changes in the late afternoon or nighttime.", please specify the wind directions, and discuss if the sources of pollutants changed.

9) Page 7, line 14: "... from 9-22 nm to 23-69 nm...", please reword it.

10) Page 8, line 29: "uparticulate" should be "particulate".

11) Page 9, line 19: "... need to confirm this.", should be "... are needed to confirm this."

12) Page 9, line 30: "stopped the growth", should be "stopped growing".

13) Page 10, line 17: "The observed concentrations of OM and NO3- largely oscillated and had no increasing trends after 21:00, although Dpg increased from 60 nm to 75 nm in one and half hours.", explain this.

[Figure]

14) Page 12, line 5: "The slope further suggests that an increase of 10 ppb in Ox likely causes an increase of 5 nm in Dpgmax.", I would suggest removing it.

15) Page 12, line 23: "When the estimated CS were compared, the averaged value was $1.8\pm2.0\times10\text{-}2$ s-1, $2.1\pm1.5\times10\text{-}2$ s-1 and $2.0\pm1.2\times10\text{-}2$ s-1. . .", the deviations are too large to provide detailed information, I would suggest removing it.

---

## Author Comment (AC1) · 11 Aug 2020

This study investigated seasonal variations of new particle formation (NPF) events in Beijing by using observations of particle size distributions and chemical compositions of aerosols and numerical model simulations. The authors found no apparent growth of new particles in winter whereas the growth of new particles to CCN size (50 or 75 nm) was often observed in summer. The three patterns of NPF events during the summertime were discussed in terms of secondary aerosol formation, evaporation of semi-volatile species, and spatial heterogeneity of NPF events. The scope of this manuscript

is well suited to ACP, and the data obtained by the authors are valuable and important to understand the mechanisms of NPF events in urban atmospheres. However, the current manuscript needs substantial revisions before the manuscript is considered as a publication of ACP as shown below.

Response: Thanks for the reviewer's comments. We will try our best to respond and revise our manuscript accordingly.

1) Page 1, Line 17: "11/27" should be revised. For example, "11 new particle formation (NPF) events out of 27 events" may be better. Other parts written similarly in the text should also be revised.

Response: Thanks for the reviewer's suggestion. We will revise this in the manuscript.

2) Page 2, Lines 21-28: The authors described what they did in this study. However, it is not clear to me which parts of this manuscript are scientifically new. There are many previous studies on NPF in Beijing and other urban areas. The authors should summarize these previous studies and describe what are well understood and what are poorly understood in Introduction. Then, the objectives of this study should be described more clearly. The sentence at Lines 18-20 (Thus far, which chemicals drive the growth.) is a point poorly understood, but I don't think the understanding on this point was improved by this study.

Response: Thanks for the reviewer's comment. The NPF events in Beijing have been widely reported (e.g., Wu et al., 2007; Yue et al., 2010; Wang et al., 2013; Kulmala et al., 2016; Chu et al., 2019). Previous studies focus mainly on the nucleation precursors (e.g., sulfuric acid, organic vapors) and the coagulation scavenge under high loadings of pre-existing particles. In this paper, we concentrated on the growth process of newly formed particles, and found that the new particles may encounter the ceiling at 20~50 nm before they grow to CCN size. The growth mechanisms can improve our understanding of the effects of NPF events on climate. More revisions will be added to better demonstrate the progress on NPF in Beijing. The sentence at Lines 18-20 will

be removed in the revised manuscript.

3) Page 4, Line 3: Equation (2) Please add descriptions on the uncertainty of this equation. Response: The uncertainty of equation (2) has been reported by Lu et al., (2019). The correlation coefficient between measured [H2SO4] and the proxy function was 0.83, and the relative error was less than 20%. These uncertainties will be added in the revised manuscript.

4) Page 4, Line 11: The SPR analysis (section 4.5) is not meaningful. It is hard to quantitatively estimate the survival fraction of new particles from this equation because the SPR values can be greater than 100% in many cases (Table 1). I think the authors may be able to calculate the loss rate of new particles during each NPF event from CS.

Response: Thanks for the comment. In the revised manuscript, we will remove the discussion of SPR into the supporting information.

5) Page 4, Line 13: Please clarify why 3 sigma was chosen.

Response: The particle number concentration follows the lognormal distribution. In the function curve, 1 sigma covers 68.27% area, 2 sigma covers 95.45% area, and 3 sigma covers 99.74% area. In this study, we use 3 sigma to represent almost all particles in this mode.

6) Page 4, Line 20: Massrequired The authors compared Massrequired with the changes in mass concentrations of organic and nitrate aerosols, but the latter is generally controlled by accumulation mode particles, not nucleation mode particles. The comparison between Massrequired (changes in aerosol mass for nucleation or Aitken mode particles) and the changes in mass concentrations of organic and nitrate aerosols (mainly controlled by accumulation mode particles) is therefore not so meaningful (in sections 4.1-4.3).

Response: Ideally, the chemical composition of nucleation mode particles would be the best to explain the growth of new particles. The data was not available in this study

unfortunately and it is still a common challenge in research community. In our previous observation using AMS, the real-time chemical composition of sub-100 nm particles has large artifacts, and AMS measurements for these small particles are not reliable. Thus, in the literature, the PM1.0 chemical composition was usually used to explain the new particles chemical information, e.g., Salimi et al. (2015) and those revised by Chu et al., (2019). In our manuscript, we combined all chemistry information to explain the growth of new particles, and the uncertainties will be added in the revised version.

7) Page 5, Lines 1-2: Please provide some brief descriptions on model setups.

Response: Thanks for the comment. We will provide brief descriptions on model setups in the revised manuscript.

8) Pages 5, Lines 8-9: Please describe on model evaluations more clearly (e.g., the degree of agreement with observations, chemical species evaluated).

Response: Thanks for the comment. We will add the model evaluations as follows: "The model results generally could meet the benchmark criteria of above four species (US-EPA, 2007), with correlations of higher than 0.51 (Table S1). The concentrations of $SO_4^{2-}$ and $NH_4^+$ had been slightly overestimated (with Normalized Mean Bia (NMBs) and 12%, 6%), while the concentrations of $NO_3^-$ and OOA were underestimated (with NMBs of -29% and -39%). Detailed evaluation results of this study could be found in Supporting Information."

9) Page 5, Lines 23-27: The unit of number concentrations in this paragraph is probably not correct.

Response: We will correct the unit in this paragraph.

10) Page 6, Lines 2-19: Please clarify why Class II was subclassified to 4 scenarios. What is the purpose of this?

Response: In Class II, we would like to distinguish scenario 1 from other types. New particles in Scenario 1 can grow to 27-48 nm and then stop growing. The stop lasted

for a few hours until the new particle signal dropped to a negligible level. This growth pattern means the new particles encounter the ceiling in the growth, and we try to interpret these events in terms of meteorology, physical and chemical properties of particulate matter. However, for other scenarios, no continuous observations can allow us to justify whether the ceiling exist or not.

11) Page 8, Line 3: the contribution of <2% Please clarify how the authors estimated this contribution. I think the authors have sulfate data observed by AMS. The data can be shown like OM and nitrate in Figures 2-4.

Response: We estimated the contribution of H2SO4 vapor to the particle growth followed the equation of Kulmala et al. (2001), i.e., R = ([H2SO4]avg/C) × 100%. where [H2SO4]avg is the average concentration of H2SO4 during the particle growth period, and C is the total concentration of condensable vapor for the particle growth, which can be calculated as described by Kulmala et al. (2001). We will add this equation in the revised manuscript. The calculation may be more accurate to reflect the contribution than the use of sulfate in PM1.0.

12) Page 8, Line 7: 13 ug m-3 Please clarify how the authors estimated this value. Did the authors consider the spread of particle size distributions? (like 3 sigma in equation (3)).

Response: We calculated the required mass follow equation (4). The averaged integral value of particle number concentration also consider the 3 sigma in the lognormal distribution.

13) Page 8, Lines 6-8: As I described above, the comparison between the required mass (13 ug m-3) and PM1.0 enhancement (15 ug m-3) is not so meaningful because the former focuses on nucleation/Aitken mode particles but the latter is usually dominated by accumulation mode particles. I think what the authors can do here is to calculate mass concentration changes for sulfate, nitrate, ammonium, and SOA and to discuss which changes are the largest during the growth periods of new particles.

Response: Thanks for the question. In the revised manuscript, we will use the change of sulfate, nitrate and OOA instead of OM, to discuss which chemical contributes largest to the particle growth. Besides, the uncertainties will be added in section 2.1.

14) Page 8, Line 7: OM can be divided into HOA (POA like) and OOA (SOA like) by using m44 and m57 signals. Only OOA can contribute to the growth of particles.

Response: Thanks for the comments. In the revised manuscript, we use the OOA data to discuss the growth of new particles.

15) Page 9, Lines 3-4: I don't agree with this authors' description. The simulated OA and nitrate cannot be used to interpret the data unless the authors evaluate the simulations with observations.

Response: We will add the model evaluations as follows: "The model results generally could meet the benchmark criteria of above four species (US-EPA, 2007), with correlations of higher than 0.51 (Table S1). The concentrations of SO42- and NH4+ had been slightly overestimated (with NMBs and 12%, 6%), while the concentrations of NO3- and OOA were underestimated (with Normalized Mean Bia (NMBs) of -29% and -39%). Detailed evaluation results of this study could be found in Supporting Information."

16) Page 10, Line 16 Delete "(ON)".

Response: Corrected.

17) Page 10, Line 28: OM can be divided to HOA and OOA as I described above.

Response: We will change OM to OOA in the main text and figures.

18) Page 11, Lines 5-6: "Repartition of the. . ." This part should be removed because no data can support this sentence.

Response: Corrected.

[Figure]

19) Page 12, Line 19 "then in", "transience": they should be corrected.

Response: We will remove "then in", and change "transience" to "the changes of season".

20) Page 12, Line 26: Section 4.5 As I described above, this section is not so meaningful and should be removed. How did the author consider the contribution of primary particles in this analysis?

Response: We removed section 4.5 into the supporting information. The contribution of primary particles, such as vehicles particles, restaurants or factories emissions will be added in the supporting information (Figure 1-2). In this study, we use a high-time resolution particle sizer, i.e., FMPS, to measure the particle number size distribution (PNSD) in 1 s time resolution. The high time resolution of FMPS can allow clearly identify the signals of newly formed particles from preexisting ambient particles, e.g., freshly emitted particles from combustion, as well as the mixing process of the different types of particles (Liu et al., 2014; Man et al., 2015; Zhu et al., 2017, 2019). During the NPF events (e.g. Figure 3), COA occasionally influences the new particles signal, and the growth of new particles was consistent with the increase in LO-OOA and MO-OOA. Therefore, we argued that we can distinguish the primary particles from new particles signal, and the growth of new particles depends largely on the condensational growth.

Reference:

Chu, B., Kerminen, V. M., Bianchi, F., Yan, C., Petäjä, T., and Kulmala, M.: Atmospheric new particle formation in China. Atmos. Chem. Phys., 19, 115-138, https://doi.org/10.5194/acp-19-115-2019, 2019.

Kulmala, M., Petaja, T., Kerminen, V., Kujansuu, J., Ruuskanen, T., Ding, A., Nie, W., Hu, M., Wang, Z., Wu, Z., Wang, L., Worsnop, D. ROn secondary new particle formation in China. Frontiers of Environmental Science & Engineering in China, 10(5):08, doi: 10.1007/s11783-016-0850-1, 2016.

Kulmala, M., Dal Maso, M., Mäkelä, M., Pirjola, L., Väkevä, M., Aalto, P., Miikkulainen, P., and Hämeri, K.: On the formation, growth and composition of nucleation mode particles, Tellus B, 53, 479–490, https://doi.org/10.1034/j.1600-0889.2001.d01-33.x, 2001.

Liu, X.H., Zhu, Y.J., Zheng, M., Gao, H.W., Yao, X.H. Production and growth of new particles during two cruise campaigns in the marginal seas of China. Atmospheric Chemistry and Physics, 2014, 14, 7941-7951.

Lu, Y., Yan, C., Fu, Y., Chen, Y., Liu, Y., Yang, G., Wang, Y., Bianchi, F., Chu, B., Zhou, Y., Yin, R., Baalbaki, R., Garmash, O., Deng, C., Wang, W., Liu, Y., Petäjä, T., Kerminen, V., Jiang, J., Kulmala, M., and Wang, L.: A proxy for atmospheric daytime gaseous sulfuric acid concentration in urban Beijing, Atmos. Chem. Phys., 19, 1971-1983, doi: 10.5194/acp-19-1971-2019, 2019.

Man, H., Zhu, Y., Ji, F., Yao, X., Lau, N. T., Li, Y. J., Lee, B.P., Chan, C. K. Comparison of daytime and nighttime new particle growth at the HKUST Supersite in Hong Kong, Environmental science & technology, 49 (12), 7170-7178, 2015.

Salimi, F., Crilley, L. R., Stevanovic, S., Ristovski, Z., Mazaheri, M., He, C. Johnson,G., Ayoko, G., Morawska, L. Insights into the growth of newly formed particles in a subtropical urban environment. Atmospheric Chemistry and Physics, 15(23), 13475-13485, 2015.

Wang, Z. B., Hu, M., Sun, J. Y., Wu, Z. J., Yue, D. L., Shen, X. J., Zhang, Y. M., Pei, X. Y., Cheng, Y. F., Wiedensohler, A. Characteristics of regional new particle formation in urban and regional background environments in the North China Plain. Atmospheric Chemistry and Physics, 13(24), 12495-12506, 2013.

Wu, Z., Hu, M., Liu, S., Wehner, B., Bauer, S., Maßling, A., Wiedensohler, A., PetäJä, T., Dal Maso, M., Kulmala, M. New particle formation in Beijing, China: Statistical analysis of a 1-year data set. Journal of Geophysical Research: Atmospheres, 112(D9), doi: 10.1029/2006JD007406, 2007.

Yue, D.L., Hu, M., Zhang, R.Y., Wang, Z.B., Zheng, J., Wu, Z.J., Wiedensohler, A., He, L.Y., Huang, X.F, Zhu, T. The roles of sulfuric acid in new particle formation and growth in the mega-city of Beijing. Atmospheric Chemistry and Physics, 10(10), 4953-4960, 2010.

Zhu, Y., Li, K., Shen, Y., Gao, Y., Liu, X., Yu, Y., Gao, H., and Yao, X.: New particle formation in the marine atmosphere during seven cruise campaigns, Atmos. Chem. Phys., 19, 89-113, doi: 10.5194/acp-19-89-2019, 2019.

Zhu, Y., Yan, C., Zhang, R., Wang, Z., Zheng, M., Gao, H., Gao, Y., and Yao, X.: Simultaneous measurements of new particle formation at 1 s time resolution at a street site and a rooftop site, Atmos. Chem. Phys., 17, 9469-9484, doi: 10.5194/acp-17-9469-2017, 2017.
* * *
[Figure]

Fig. 1. Contour plot of traffic emissions (a) and the raw FMPS data of the vehicle spikes (b)

[Figure]

**Fig. 2.** Fresh industrial emissions with high SO2 (12:30-15:00) and the cooking emissions with increased cooking OA (COA, 18:20-21:00)

**Fig. 3.** NPF event and the variations of hydrocarbon-like OA (HOA), cooking OA (COA), less oxidized oxygenated OA (LO-OOA) and more oxidized oxygenated OA (MO-OOA) on June 18, 2014

---

## Author Comment (AC2) · 11 Aug 2020

This study investigated patterns and characteristics of atmospheric new-particle formation events in Beijing. The authors categorized these observed events into three classes based on the number size distributions of the newly-formed particles. Further, by combing the size distribution with the speciation of measured or modelled gas and particle-phase pollutants, the authors discussed the contribution of organic and inorganic compounds to particle growth during different type of events.

[Figure]

The manuscript focus on the topic of new-particle formation in the urban atmosphere, trying to address critical questions that whether or not the newly-formed particles can grow to the CCN size, and what conditions/species control the grow process. The scope of the manuscript is thus suitable for ACP, and the data the authors presented are ample and interesting. However, the interpretation of some key results is questionable and leads to unrigorous conclusions. Major revisions and improvements are needed before this manuscript can be considered for publication as an ACP paper.

Response: The authors thank the reviewer's comments and try our best to respond and revise our manuscript accordingly.

Major comments:

1) The authors defined three classes. "Class I was characterized by no apparent particle growth" makes sense, this class might indicate either a lack of supersaturated condensable vapors so that particles don't grow, or a too high condensation sink so that small particles don't survive. But is there a better way to classify the rest events? Particles are larger than 50 (or 75) nm doesn't necessarily mean they are good CCNs; and there are so many factors (chemical, physical or meteorological) that can determine whether or not the particles grow over 50 (or 75) nm. Classifying the events just based on the "cut-off" size doesn't really help modelers or lab experimentalists to understand the real atmosphere. Please justify the classification or improve it.

Response: Thanks for your comment. Previous studies in various atmospheric environments showed that particles' ability to act as CCN is largely determined by aerosol size at normal ambient super-saturation when the size of atmospheric particles is smaller than 60-80 nm (Dusek et al., 2006; Li et al., 2015; Zhu et al., 2019). However, chemical composition, particle mixing states, meteorological conditions, etc., may affect the activation of particles to CCN with diameters beyond 60-80 nm. In our previous studies, we found that newly formed particles sometimes encounter a growth threshold around 50 nm, which can't be explained by the meteorological conditions (Man et al., 2015, Zhu

et al., 2014, 2017, 2019). Therefore, we use the particle "cut-off" size as classification criteria. In the revision, we will add more justification and improve it to better service other researchers.

2) From the surface plot of these NPF events (e.g. Fig 2a, e; Fig 3a, e; Fig 4a), I don't see any significant band of pre-existing particles. Were these events all observed in very clean days? Or is it because the linear color scale veil the background particles? Please do change to the log color scales.

Response: Thanks for your suggestion. We have changed the liner color scale to the log color scale (see the revised manuscript), and the pre-existing particles are more obvious in the revised figures.

3) The author stated that many growth events lasted for over 10 hrs or even a whole day. Was there any primary emission mixing with the newly-formed particles, e.g. from vehicles, restaurants or factories? Is it true that there was only condensational growth without mixing during the whole period? Please discuss this and also show O:C from the AMS measurement to verify the statement.

Response: Thanks for your comment. We use a high-time resolution particle sizer, i.e., FMPS, to measure the particle number size distribution (PNSD) in 1 s time resolution. The high time resolution of FMPS can allow clearly identify the signals of newly formed particles from preexisting ambient particles, e.g., freshly emitted particles from combustion, as well as the mixing process of the different types of particles (Liu et al., 2014; Man et al., 2015; Zhu et al., 2017, 2019). According to our previous studies (e.g. Zhu et al., 2017, ACP) and the review paper by Tuan et al. (2015), the PNSD of traffic emissions are characterized by two peaks, i.e., about 16 nm and 30 nm, and intermittently lasts a few seconds or minutes (Figure 1). Figure 2 (will be added in new supporting information) shows the fresh industrial emissions and/or the cooking emissions. During the NPF events (e.g. Figure 3), COA occasionally influences the new particles signal, and the growth of new particles was consistent with the increase in LO-OOA and MO-

OOA. Therefore, we argued that the growth of new particles depends largely on the condensational growth.

4) About the AMS measurements, the sampling site is 8 km away from Peking University, how long does it take for an air parcel transport from one site to the other? Roughly one hour maybe? How well does the AMS result represent the particle composition at Peking University? I think this question need to be better addressed in order to discuss the spatial heterogeneity.

Response: NPF events are regional occurred. In our study, the average duration of NPF events was 10 hrs., and the wind speed was approximately 4-6 m/s. There was only half hour delay for air parcels sweeping from one site deployed the AMS to another site deployed the FMPS. Under such wind speed, the chemical components in PM1.0 measured by the AMS mainly represented regional characteristics. However, local emissions may also contribute to then to some extent. In our previous study (He et al., 2002), the chemical composition of PM2.5 at two sampling sites in the zone of Beijing were generally consistent. The uncertainty will be added in the revised manuscript.

5) AMS measured the bulk $PM_{1.0}$, how well does the chemical composition in $PM_{1.0}$ represent the species drive the sub-100 nm particle growth? Were there any aerodynamic diameters measured by AMS at the same time? Discuss more about the uncertainty here.

Response: Thanks for the question. Ideally, the real-time chemical composition of sub-100 nm particles would be the best to explain the growth of new particles. The data was not available in this study unfortunately. In our previous observation using AMS, the real-time chemical compositions of sub-100 nm particles have large artifacts. AMS measurements for these small particles may be not reliable. We thereby combined all chemistry information to explain the growth of new particles. In the revised manuscript, the uncertainties will be added in section 2.1.

6) Page 4, line 20, Equation 4, I don't find the exact same equation in the references

the authors cited here. Using averaged particle number concentration over the whole growth period can bring in large uncertainties. Newly-formed particle is prone to coagulation loss; this means particle number concentration at D_pg1 will be much higher than that at D_pg2, and the Mass required will be overestimated. Please justify the equation, or calculate the particle mass concentration for each FMPS scan and also take the coagulation into account.

Response: Agree. A more accurate calculation should be adopted to gain the required mass for particle growth. During the new particle growth period, the number concentration may decrease due to the dry deposition, diffusion and dilution effects, and the particle coagulation. In the revised manuscript, we will calculate the range of required mass according to the integral particle number concentration at Dpg,1 (N1) and Dpg,2 (N2). The formula will change to

Massrequried = 4/3 [(Dpg2/2)3 - (Dpg1/2)3] * N1 *  ∼4/3 [(Dpg2/2)3 - (Dpg1/2)3] * N2 * 

7) Was there a special reason to sum up O3 and NO2 instead of discussing them separately? NO2 is not always associated with O3, it could come from primary emission such as vehicle exhaust.

Response: In China, previous study showed that the primary on-road vehicular NO2/NOx ratio was less than 2%, and NO was the main exhaust gas (Yao et al., 2005). Heavy duty vehicles are allowed to enter urban areas in Beijing only after 20:00 but not in daytime. Considering the NO-titration of O3 (O3+NO→NO2+O2), we use the sum of O3 and NO2 to represent the oxidizing capacity.

8) The authors briefly mentioned seasonal variation, but didn't dive into the details in, for example, wintertime events. A recent study (Wang et al., 2020, Nature 581 (7807), 184-189) show that NH4NO3 could help the newly-formed particles grow and survive in winter. So, it's intriguing to know that the authors observed that the newly formed particles didn't grow during wintertime events, but it would be more important

to understand why they didn't. Was it because of a lack of supersaturated condensable vapors, or a too high condensation sink?

Response: Wang et al. (2020) performed the experiments in the CLOUD chamber with scrupulous cleanliness and minimal contamination, and found NH4NO3 can drive the newly-formed particles grow to the larger sizes. However, NH4NO3 were less likely formed during winter daytime NPF period in Beijing because of low products of HNO3 and NH3 in ambient air. In the chamber study reported by Guo et al. (2020, PNAS, 117, 7, 3427-3432), new particles grew rapidly to about 50 nm in the clean chamber, but they grew much slowly in the polluted chamber, in which the sizes of new particles less than 20 nm (Figure 1 in Guo's paper). Based on Guo's study, the lack of growth of new particle during wintertime was due to the high condensation sink. This will be added in the revision.

9) Please discuss more about the model uncertainty, sensitivity test, etc.

Response: In the revised manuscript, we have added the model uncertainty and sensitivity test in section 3.2, The model results generally could meet the benchmark criteria of above four species (US-EPA, 2007), with correlations of higher than 0.51 (Table S1). The concentrations of SO42- and NH4+ had been slightly overestimated (with Normalized Mean Bia (NMBs) and 12%, 6%), while the concentrations of NO3- and OOA were underestimated (with NMBs of -29% and -39%).

10) The authors used many sentences describing the particle growth/shrinkage processes vs time, e.g. "However, the shrinkage occurred as early as 15:20-17:20 on 11 June.". Yet these sentences contain very limited information. I would suggest the authors go through the whole manuscript and reword these sentences by discussing more deeply about the environmental conditions or the causes of these different types of growth.

Response: Thanks for the comment. In the revised manuscript, the relationship between the meteorological conditions and the new particle growth types will be discussed in detail.

Minor comments: 1) Page 1, line 17: "11/27" (and hereinafter) should be something like "11 out of 27".

Response: Corrected.

2) Page 1, line 22: ": : : in the remaining NPF events", please add the number here.

Response: Corrected.

3) Page 2, line 26: "survival probability ratios", "survival probability" would be better.

Response: Corrected.

4) Page 3, line 9: should the coefficient be size-dependent?

Response: The coefficient was dependent on the particle number concentration, no the particle sizes.

5) Page 3, line 13: "During the other observational periods: : :", please specify the date.

Response: Corrected.

6) Page 6, line 8: "to a negligible level; in Scenario 2", the semicolon should be period.

Response: Corrected.

7) Page 6, line 10: ": : : may not represent two NPF events occurring in one day.", why? They look very much like two events.

Response: In the revised manuscript, this sentence will be modified to "This phenomenon can be explained by the spatial heterogeneity of NPF, or it can represent two NPF events that occurred in one day."

8) Page 6, line 14: ". . . associated with wind direction changes in the late afternoon or nighttime.", please specify the wind directions, and discuss if the sources of pollutants

changed.

Response: We will add the wind speeds and wind directions in the supporting information.

9) Page 7, line 14: ". . . from 9-22 nm to 23-69 nm. . .", please reword it.

Response: This sentence will be modified to "the initial Dpg of newly formed particles were 9-22 nm, and they grew to 23-69 nm in the daytime".

10) Page 8, line 29: "uparticulate" should be "particulate".

Response: Corrected.

11) Page 9, line 19: ". . . need to confirm this.", should be ". . . are needed to confirm this."

Response: Corrected.

12) Page 9, line 30: "stopped the growth", should be "stopped growing".

Response: Corrected.

13) Page 10, line 17: "The observed concentrations of OM and NO3- largely oscillated and had no increasing trends after 21:00, although Dpg increased from 60 nm to 75 nm in one and half hours.", explain this.

Response: In the revised manuscript, we will change OM to OOA, and discuss the variations of OOA during particles growth. In this case (Figure 4), OOA reaches its maximum at 22:20 and is consistent with the increasing of Dpg.

14) Page 12, line 5: "The slope further suggests that an increase of 10 ppb in Ox likely causes an increase of 5 nm in Dpgmax.", I would suggest removing it.

Response: We will remove this sentence in the revised manuscript.

15) Page 12, line 23: "When the estimated CS were compared, the averaged value

was 1.8±2.0×10-2 s-1, 2.1±1.5×10-2 s-1 and 2.0±1.2×10-2 s-1. . .", the deviations are too large to provide detailed information, I would suggest removing it.

Response: We will remove this sentence in the revised manuscript.

Reference:

Dusek, U., Frank, G. P., Hildebrandt, L., Curtius, J., Schneider, J., Walter, S., Chand, D., Drewnick, F., Hings, S., Jung, D., Borrmann, S., Andreae, M. O. Size Matters More Than Chemistry for Cloud-Nucleating Ability of Aerosol Particles. Science, 312, 1375-1378, 2006.

Guo, S., Hu, M., Peng, J., Wu, Z., Zamora, M. L., Shang, D., Du, Z., Zheng, J., Fang, X., Tang, R., Wu, Y., Zeng, L., Shuai, S., Zhang, W., Wang, Y., Ji, Y., Zhang, A., Wang, W., Zhang, F., Zhao, J., Gong, X., Wang, C., Molina, M., Zhang, R. Remarkable nucleation and growth of ultrafine particles from vehicular exhaust. Proceedings of the National Academy of Sciences of the United States of America, 117(7), 3427-3432, 2020.

Li, K., Zhu, Y.J., Gao, H.W., Yao, X.H., A comparative study of cloud condensation nuclei measured between non-heating and heating periods at a suburb site of Qingdao in the North China. Atmospheric Environment, 112, 40-53, 2015.

Liu, X.H., Zhu, Y.J., Zheng, M., Gao, H.W., Yao, X.H. Production and growth of new particles during two cruise campaigns in the marginal seas of China. Atmospheric Chemistry and Physics, 2014, 14, 7941-7951.

Ma, N., Zhao, C., Tao, J., Wu, Z., Kecorius, S., Wang, Z., Größ, J., Liu, H., Bian, Y., Kuang, Y., Teich, M., Spindler, G., Müller, K., van Pinxteren, D., Herrmann, H., Hu, M., and Wiedensohler, A.: Variation of CCN activity during new particle formation events in the North China Plain, Atmos. Chem. Phys., 16, 8593-8607, doi: 10.5194/acp-16-8593-2016, 2016.

Man, H., Zhu, Y., Ji, F., Yao, X., Lau, N. T., Li, Y. J., Lee, B.P., Chan, C. K. Comparison of daytime and nighttime new particle growth at the HKUST Supersite in Hong Kong,

Environmental science & technology, 49 (12), 7170-7178, 2015.

Tuan, V., Delgadosaborit, J. M., & Harrison, R. M. Review: Particle number size distributions from seven major sources and implications for source apportionment studies. Atmospheric Environment" 114-132, 2015.

Yao, X. H., Lau, N. T., Fang, M., Chan, C. K., Real-time observation of the transformation of ultrafine atmospheric particle modes. Aerosol Science and Technology, 2005, 39(9), 831-841.

Zhu, Y., Li, K., Shen, Y., Gao, Y., Liu, X., Yu, Y., Gao, H., and Yao, X.: New particle formation in the marine atmosphere during seven cruise campaigns, Atmos. Chem. Phys., 19, 89-113, doi: 10.5194/acp-19-89-2019, 2019.

Zhu, Y., Yan, C., Zhang, R., Wang, Z., Zheng, M., Gao, H., Gao, Y., and Yao, X.: Simultaneous measurements of new particle formation at 1 s time resolution at a street site and a rooftop site, Atmos. Chem. Phys., 17, 9469-9484, doi: 10.5194/acp-17-9469-2017, 2017.
* * *
[Figure]

Fig. 1. Contour plot of traffic emissions (a) and the raw FMPS data of the vehicle spikes (b)

**Fig. 2.** Fresh industrial emissions with high SO2 (12:30-15:00) and the cooking emissions with increased cooking OA (COA, 18:20-21:00)

**Fig. 3.** NPF event and the variations of hydrocarbon-like OA (HOA), cooking OA (COA), less oxidized oxygenated OA (LO-OOA) and more oxidized oxygenated OA (MO-OOA) on June 18, 2014

**Fig. 4.** NPF event that occurred on 27 June 2014 ((a) contour plot of the particle number concentration; (b) time series of the observed mixing ratios of SO2, NO2+O3 (c) time series of observed OOA, NO3- and N

---

## Author Response (AR1)

*This study investigated patterns and characteristics of atmospheric new-particle formation events in Beijing. The authors categorized these observed events into three classes based on the number size distributions of the newly-formed particles. Further, by combing the size distribution with the speciation of measured or modelled gas and particle-phase pollutants, the authors discussed the contribution of organic and inorganic compounds to particle growth during different type of events.*

*The manuscript focus on the topic of new-particle formation in the urban atmosphere, trying to address critical questions that whether or not the newly-formed particles can grow to the CCN size, and what conditions/species control the grow process. The scope of the manuscript is thus suitable for ACP, and the data the authors presented are ample and interesting. However, the interpretation of some key results is questionable and leads to unrigorous conclusions. Major revisions and improvements are needed before this manuscript can be considered for publication as an ACP paper.*

**Response:** The authors thank the reviewer's comments and try our best to respond and revise our manuscript accordingly.

*Major comments:*

*1)The authors defined three classes. "Class I was characterized by no apparent particle growth" makes sense, this class might indicate either a lack of supersaturated condensable vapors so that particles don't grow, or a too high condensation sink so that small particles don't survive. But is there a better way to classify the rest events? Particles are larger than 50 (or 75) nm doesn't necessarily mean they are good CCNs; and there are so many factors (chemical, physical or meteorological) that can determine whether or not the particles grow over 50 (or 75) nm. Classifying the events just based on the "cut-off" size doesn't really help modelers or lab experimentalists to understand the real atmosphere. Please justify the classification or improve it.*

**Response:** When the size of atmospheric particles is smaller than 60 nm (Dusek et al., 2006), the CCN activation of aerosol particles at normal ambient super-saturation has been reported to be determined solely by particle size. In addition to particle size, various factors such as chemical composition, particle mixing states, and meteorological conditions may also largely affect CCN activation of aerosols with $D_{pg}$ beyond 70 nm (Ma et al., 2016; Rose et al., 2017; Lee et al., 2019). Although new particles in Class III can grow to the CCN size, the CCN activation of grown new particles has been reported to vary case by case (Wiedensohler et al., 2009; Yue et al., 2011; Li et al., 2015; Ma et al., 2016). This has been added in the revision (Page 8, Lines 23-27).

As presented in the origin manuscript, currently in Page 2, lines 15-21 of the revision, "Nevertheless, reported observations have also shown that newly formed particles with diameters less than 40–50 nm can be activated as CCN only under high supersaturation (SS), such as >0.6% (Li et al., 2015; Ma et al., 2016). When newly formed particles grow with the geometric median diameter to larger than 70 nm, they significantly contribute to the CCN population at SS≤0.2% (Wiedensohler et al., 2009; Yue et al., 2011; Li et al., 2015; Ma et al., 2016; Zhu et al., 2019). In addition, field observations have also shown that in most NPF events, the maximum geometric median diameter ($D_{pgmax}$) of newly grown particles is less than 40–50 nm before new particle signals drop to a negligible level (Zhu et al., 2014, Liu et al., 2014; Man et al., 2015; Zhu et al., 2017; Yu et al., 2019)." It is clear that the growth of newly formed particles encountered a ceiling in size less than 40-50 nm during most of NPF events. This was also true in Scenario 1 of Class II in Beijing. The ceiling prevented newly formed particles from growing to the CCN size in Scenario 1 (This has been added in Page 8, lines 12-13). What causes the ceiling in size less than 40-50 nm and what causes newly formed particles growing over the ceiling are crucially important for modelers or lab experimentalists to explore the true contribution of NPF to the population of CCN.

In fact, 50 nm has been widely used a threshold to judge grown new particles as CCN in high supersaturation in review literature (Kerminen et al., 2018). For the threshold of 70 nm, we added "Similar definitions are applied for the SP of grown new particles with $D_{pg}$ reaching over 70 nm, in which grown new particles can be activated as CCN with highly variable activation efficiencies." Page 6, lines 2-3 in the revision.

*2) From the surface plot of these NPF events (e.g. Fig 2a, e; Fig 3a, e; Fig 4a), I don't see any significant band of pre-existing particles. Were these events all observed in very clean days? Or is it because the linear color scale veil the background particles? Please do change to the log color scales.*

**Response:** Thanks for your suggestion. We have changed the liner color scale to the log color scale (see the revised manuscript), and the pre-existing particles are more obvious in the revised figures.

*3) The author stated that many growth events lasted for over 10 hrs or even a whole day. Was there any primary emission mixing with the newly-formed particles, e.g. from vehicles, restaurants or factories? Is it true that there was only condensational growth without mixing during the whole period? Please discuss this and also show O:C from the AMS measurement to verify the statement.*

**Response:** The questions reflect the common challenge when a SCANNING particle sizer

operating in LOW time resolution in MINUTES is used for sampling and studying NPF events. However, it is not an issue when a PARALLELING particle sizer operating in high time resolution in ONE SECOND is used for the same target study.

We used a high-time resolution PARALLELING particle sizer, i.e., FMPS, to measure the particle number size distribution (PNSD) in 1 s time resolution. The high time resolution of FMPS can allow clearly identify the signals of newly formed particles from preexisting ambient particles, e.g., freshly emitted particles from combustion, as well as the mixing process of the different types of particles (Liu et al., 2014; Man et al., 2015; Zhu et al., 2017, 2019).

According to our previous studies (e.g. Zhu et al., 2017, ACP) and the review paper by Tuan et al. (2015), the PNSD of traffic emissions are characterized by two peaks, i.e., about 16 nm and 30 nm, and intermittently lasts a few seconds or minutes (Figure R1). Figure R2 (added in new supporting information) shows the fresh industrial emissions associated with $SO_2$ and/or the cooking emissions associated with increased COA at 18:00-20:00. It is clear that the dominant modes of particles from traffic emissions, industrial emissions and cooking emissions occurred at ~20 nm, ~30 nm and ~40 nm, respectively, and the domain mode size were quietly stable in the study period. Their contributions were important during non-NPF periods, which were not the focus of this study. On the roof sampling site, their contributions to the observed particle concentrations during the initial few hours of NPF were generally negligible in presence of wind speeds of 4-6 m $s^{-1}$, except a few occasional spikes lasting in minutes. These spikes were excluded in calculating NMINP. In the revision, we add "Note that a few spikes of several minutes were occasionally observed and were excluded to calculate NMINP because they may reflect primary particles from localized sources (Liu et al., 2014; Zhu et al., 2017).". The same is true in calculating the growth rate of particles, shrinkage rate and $D_{pgmax}$. Additionally, their influences can also be ignored in studying the growth of newly formed particles when the particles grew over 50 nm.

During the NPF events (e.g. Figure R3), COA occasionally influences the new particles signal, and the growth of new particles was consistent with the increase in MO-OOA and LO-OOA. Therefore, we argued that the growth of new particles depends largely on the condensational growth.

We prefer to add the argument in Supporting Information since the challenge is not immediately related to the samplings and associated analyses presented in the text.

[Figure]

Figure R1 Contour plot of particle number concentrations at the roadside site with spikes from traffic emissions.

[Figure]

Figure R2 Fresh industrial emissions associated with high SO$_2$ (12:30-15:00) and cooking emissions with increased cooking OA (COA, 18:20-21:00).

[Figure]

Figure R3 NPF event and variation in hydrocarbon-like OA (HOA), cooking OA (COA), less oxidized oxygenated OA (LO-OOA) and more oxidized oxygenated OA (MO-OOA) on 18 June 2014.

*4) About the AMS measurements, the sampling site is 8 km away from Peking University, how long does it take for an air parcel transport from one site to the other? Roughly one hour maybe? How well does the AMS result represent the particle composition at Peking University? I think this question need to be better addressed in order to discuss the spatial heterogeneity.*

**Response:** In the revision, page 4, lines 15-21, we add "During NPF events at wind speeds of 4–6 m s$^{-1}$, a half-hour delay may occur for air parcels sweeping from one site with the FMPS deployed to another site with the AMS deployed. For NPF events with durations over several hours, the events were expected to occur regionally (Kerminen et al., 2018; Chu et al., 2019). Thus, it is reasonable to interpret the cumulative growth of newly formed particles within several hours, measured by the FMPS, by using the net simultaneous change in concentrations of chemical species, measured by the AMS. Additionally, He et al. (2001) reported that the chemical composition of PM$_{2.5}$ was reasonably homogeneous in the two sampling site zones."

*5) AMS measured the bulk PM_1.0, how well does the chemical composition in PM_1.0 represent the species drive the sub-100 nm particle growth? Were there any aerodynamic diameters measured by AMS at the same time? Discuss more about the uncertainty here.*

**Response:** In the revision, page 4, lines 9-13, we add "The chemical composition of $PM_{1.0}$ measured by AMS has been widely used to interpret NPF events in the literature (Wiedensohler et al., 2009; Zhang et al., 2014; Man et al., 2015; Du et al., 2017; Rodelas et al., 2019; Kanawade et al., 2020) and was also used in this study. Low loadings of particulate chemical species in nanometer size ranges do not facilitate accurate measurement of their concentrations therein. However, the chemical composition of nanometer particles may differ from those of $PM_{1.0}$ (Ehn et al., 2014; Wu et al., 2016)."

In the revision, page 10, lines 13- 24, we add "As mentioned above, the growth of newly formed particles is mainly attributed to sulfuric acid, ammonium nitrate, and secondary organic compounds (Wiedensohler et al., 2009; Riipinen et al., 2011; Zhang et al., 2012; Ehn et al., 2014; Man et al., 2015; Wang et al., 2015; Burkart et al., 2017; Lee et al., 2019; Wang et al., 2020). We therefore explore their respective contributions as follows. First, we calculated the contribution of sulfuric acid to the growth based on the observed mixing ratio of $SO_2$ and Equations 2–4. Second, we examined whether $NH_4NO_3$ freshly formed in $PM_{1.0}$ or $PM_{2.5}$ during the particle growth period. In case of no $NH_4NO_3$ formation, its contribution would not be expected. This is because an even higher product of $HNO_{3gas}*NH_{3gas}$ is required to overcome the kelvin effect and form $NH_4NO_3$ in nucleation mode and Aitken mode particles. Thus, the growth unexplained by sulfuric acid should be mainly contributed by SOA. Third, in case of $NH_4NO_3$ formation, we compared the net increase in $NH_4NO_3$ with that in SOA. It is noteworthy that this approach is limited by the uncertainty in explaining the growth because the ratios of increased $NH_4NO_3$ over increased SOA in $PM_{1.0}$ or $PM_{2.5}$ may not be the same as the ratios in nucleation mode and Aitken mode particles. In this case, the required mass of $NH_4NO_3$ or SOA to the growth was also estimated and compared with their respective net increases to facilitate the analysis."

*6) Page 4, line 20, Equation 4, I don't find the exact same equation in the references the authors cited here. Using averaged particle number concentration over the whole growth period can bring in large uncertainties. Newly-formed particle is prone to coagulation loss; this means particle number concentration at D_pg1 will be much higher than that at D_pg2, and the Mass required*

*will be overestimated. Please justify the equation, or calculate the particle mass concentration for each FMPS scan and also take the coagulation into account.*

**Response:** Agree. The part has been revised as "The amount of chemical species required to grow new particles from $D_{pg1}$ to $D_{pg2}$ ($Mass_{requried}$) is approximately estimated as follows:

$$Mass_{requried} = 4/3\pi \, [(D_{pg2}/2)^3 - (D_{pg1}/2)^3] * N * \rho \qquad (5)$$

$\rho$ is the density, which is assumed as 1.5 µg m$^{-3}$ for OOA (or SOA) and 1.7 µg m$^{-3}$ for NH$_4$NO$_3$, respectively. Considering that the particle number concentration may decrease because of the dry deposition, diffusion and dilution effects, and particle coagulation, N represents the integral value of new particle number concentrations with the geometric median diameter of new particles from $D_{pg2-3\sigma}$ to $D_{pg2+3\sigma}$. The approximate value may overestimate the required amount because particle–particle coagulation has not been deducted."

However, inclusion of the particle-particle coagulation needs the approximation on the size of particles coagulated from two particles. The approximation would introduce additional uncertainty. We hope that the reviewer can agree on this point.

*7) Was there a special reason to sum up $O_3$ and $NO_2$ instead of discussing them separately? $NO_2$ is not always associated with $O_3$, it could come from primary emission such as vehicle exhaust.*

**Response:** In China, our previous study showed that the primary on-road vehicular NO$_2$/NO$_x$ ratio was less than 2%, and NO was the main exhaust gas (Yao et al., 2005). Heavy duty vehicles are allowed to enter urban areas in Beijing only after 20:00 but not in daytime. Considering the NO-titration of O$_3$ (O$_3$+NO→NO$_2$+O$_2$), we use the sum of O$_3$ and NO$_2$ to represent the oxidizing capacity.

*8) The authors briefly mentioned seasonal variation, but didn't dive into the details in, for example, wintertime events. A recent study (Wang et al., 2020, Nature 581 (7807), 184-189) show that $NH_4NO_3$ could help the newly-formed particles grow and survive in winter. So, it's intriguing to know that the authors observed that the newly formed particles didn't grow during wintertime events, but it would be more important to understand why they didn't. Was it because of a lack of supersaturated condensable vapors, or a too high condensation sink?*

**Response:** Wang et al. (2020) performed the experiments in the CLOUD chamber with scrupulous cleanliness and minimal contamination, and found NH$_4$NO$_3$ can drive the newly-formed particles grow to the larger sizes. The reference has been cited in the revision. However, the new finding

needs more field measurements to confirm. In our previous study (Yu et al., 2016), $NH_4NO_3$ can be formed through artifact reactions when ambient pressure was reduced largely in sampling.

Lack of apparent particle growth during NPF events in December was observed simultaneously at two neighbor sites using two identic FMPS, i.e., a roof site and a roadside site (Zhu et al., 2017), allows us having a strong confidence on the observations. Lack of apparent particle growth was also simultaneously observed during parts of NPF events in April at the two sides using the same methods. The repeated results further confirm the finding. Unfortunately, we had no simultaneous gaseous $HNO_3$ and $NH_3$ together with their particulate partners to confirm this. Thus, we cannot speculate more on this issue.

In the chamber study reported by Guo et al. (2020, PNAS, 117, 7, 3427-3432), new particles grew rapidly to about 50 nm in a clean chamber with preexisting particles removed, but they grew much slowly in the polluted air, in which the sizes of new particles grew less than 20 nm (Figure 1 in Guo's paper). Based on Guo's study, the lack of growth of new particle during wintertime was partially due to the high condensation sink. The reference has also been cited in the revision.

*9) Please discuss more about the model uncertainty, sensitivity test, etc.*

**Response:** In the revision (Page 7, lines 4-10), we have added "During the study period, the model results generally met the benchmark criteria of the above four species (US-EPA, 2007), with correlations between modeled and measured values larger than 0.57 (Table S1). The modeled concentrations of $NH_4^+$ reasonably agree with the observations with a normalized mean bias (NMB) of 6%. The NMB slightly increased up to 12% for the modeled concentrations of $SO_4^{2-}$. The modeled values of $NO_3^-$ and SOA were underestimated with NMBs of -29% and -39%, respectively. Underestimation of SOA is a common weakness of the model simulation because a fraction of SOA precursors are not included, and some key formation pathways of SOA may still be missing in current air quality models (Baek et al., 2011; Knote et al., 2014)."

*10) The authors used many sentences describing the particle growth/shrinkage processes vs time, e.g. "However, the shrinkage occurred as early as 15:20-17:20 on 11 June.". Yet these sentences contain very limited information. I would suggest the authors go through the whole manuscript and reword these sentences by discussing more deeply about the environmental conditions or the causes of these different types of growth.*

**Response:** The detailed analysis was presented in Section 4.3. In the revision (Page 13, lines 19-22), we added "During this period of shrinkage, the observed mixing ratio of $O_x$ largely decreased from 130 ppb to 80 ppb, and the observed OOA decreased from 16.2 µg m$^{-3}$ to 11.4 µg

m$^{-3}$ (Fig. 4b, c). However, the concentrations of NH$_4^+$ were almost constant. Repartition of the semivolatile SOA in gas and particle phases was hypothesized to cause the evaporation of semivolatile SOA to the gas phase." Additionally, we had revised our analysis through the manuscript.

*Minor comments:*
*1) Page 1, line 17: "11/27" (and hereinafter) should be something like "11 out of 27".*

**Response:** Corrected.

*2) Page 1, line 22: "... in the remaining NPF events", please add the number here.*

**Response:** Corrected.

*3) Page 2, line 26: "survival probability ratios", "survival probability" would be better.*

**Response:** Corrected.

*4) Page 3, line 9: should the coefficient be size-dependent?*

**Response:** We follow an empirical correction procedure for size distribution data reported by FMPS proposed by we Zimmerman (2015). The coefficient was size in-dependent.

***5) Page 3, line 13: "During the other observational periods…", please specify the date.***

**Response:** Corrected.

*6) Page 6, line 8: "to a negligible level; in Scenario 2", the semicolon should be period.*

**Response:** Corrected.

*7) Page 6, line 10: "... may not represent two NPF events occurring in one day.", why?*
*They look very much like two events.*

**Response:** The sentence has been removed in the revision to avoid confusion.

*8) Page 6, line 14: "... associated with wind direction changes in the late afternoon or nighttime.", please specify the wind directions, and discuss if the sources of pollutants changed.*

**Response:** The sentence has been removed in the text, but **t**he wind speeds and wind directions has been added in the supporting information.

*9) Page 7, line 14: "… from 9-22 nm to 23-69 nm…", please reword it.*

**Response:** Done.

*10) Page 8, line 29: "uparticulate" should be "particulate".*

**Response:** Corrected.

*11) Page 9, line 19: "… need to confirm this.", should be "… are needed to confirm this."*

**Response:** Corrected.

*12) Page 9, line 30: "stopped the growth", should be "stopped growing".*

**Response:** Corrected.

*13) Page 10, line 17: "The observed concentrations of OM and $NO_3^-$ largely oscillated and had no increasing trends after 21:00, although $D_{pg}$ increased from 60 nm to 75 nm in one and half hours.", explain this.*

**Response:** In the revised manuscript, we used OOA instead of OM. The part has been revised as (also show in Figure R4) "The observed concentrations of OOA (left axis) and $NO_3^-$ (right axis) rapidly increased from 18:00 to 22:20, with the former being approximately four times larger than the latter. The required amount of $NH_4NO_3$ for particle growth during the period was estimated to be 5.3 µg m$^{-3}$, while the net increase in $NH_4NO_3$ was 1.6 µg m$^{-3}$. SOA may dominate the growth of new particles."

[Figure]

Figure R4 NPF event that occurred on 27 June 2014 ((a) contour plot of the particle number concentration; (b) time series of the observed mixing ratios of $SO_2$, $NO_2+O_3$ (c) time series of observed OOA, $NO_3^-$ and $NH_4^+$ in $PM_{1.0}$).

*14) Page 12, line 5: "The slope further suggests that an increase of 10 ppb in $O_x$ likely causes an increase of 5 nm in $D_{pgmax}$.", I would suggest removing it.*

**Response:** Delete.

*15) Page 12, line 23: "When the estimated CS were compared, the averaged value was $1.8\pm2.0\times10^{-2}$ $s^{-1}$, $2.1\pm1.5\times10^{-2}$ $s^{-1}$ and $2.0\pm1.2\times10^{-2}$ $s^{-1}$...", the deviations are too large to provide detailed information, I would suggest removing it.*

**Response:** Delete.

**Response:** In the revision (Page 2, line 24 – Page 3, line 9), we added "With distinctive particle growth patterns being widely reported, Beijing is an ideal area for studying the growth of newly formed particles (Wehner et al., 2004; Wu et al., 2007, 2016; Wiedensohler et al., 2009; Yue et al., 2010; Matsui et al., 2011; Wang et al., 2013; Guo et al., 2014, 2020; Du et al., 2017; Zhu et al., 2017; Brean et al., 2019; Chen et al., 2019). For instance, as the first study of NPF events in Beijing, Wehner et al. (2004) reported a small growth rate (~1 nm h$^{-1}$) of newly formed particles during 25 days from March 05 to April 18, 2004. Such small growth rates are unlikely to facilitate the growth

of particles to reach CCN sizes prior to removal from ambient air because of the large coagulation loss in Beijing (Kulmala and Kerminen, 2008; Kulmala et al., 2016; Chu et al., 2019; Guo et al., 2020). Similar to this finding, no apparent growth of newly formed particles with the $D_{pgmax}$ of approximately 10 nm always occurred in December 2011 at the same campus in Beijing (Zhu et al., 2017). In contrast, the growth of newly formed particles to CCN size and even larger has also been observed in Beijing (Wu et al., 2007; Wiedensohler et al., 2009; Yue et al., 2010; Wang et al., 2013; Guo et al., 2014; Wu et al., 2016). The patterns of particle growth have not been well characterized. Nevertheless, sulfuric acid and/or organic vapors have been proposed to drive particle growth in different NPF events (Wiedensohler et al., 2009; Yue et al., 2010; Wu et al., 2016). Recently, the formation of $NH_4NO_3$ has been proposed as a driver of the rapid growth of newly formed particles in field studies and chamber experiments (Zhu et al., 2104; Man et al., 2015; Wang et al., 2020). The role of $NH_4NO_3$ in the growth of newly formed particles in Beijing remains poorly understood. Matsui et al. (2011) and Chen et al. (2019) simulated NPF and the growth of newly formed particles based on observations, but the modeling results were explained with large uncertainties."

The sentence at Lines 18-20 of the origin manuscript has been revised as "Thus, it is important to characterize NPF events, based on the $D_{pgmax}$ of grown new particles, and to explore the chemicals driving the growth of newly formed particles with $D_{pgmax}$ greater than 70 nm". Moreover, we add the logic flow in analyzing which chemicals drive the growth in newly formed particles beyond 70 nm. Please see our response to Question 6.

*3) Page 4, Line 3: Equation (2)*
*Please add descriptions on the uncertainty of this equation.*
**Response:** In the revision (Page 5, lines 8-9), we added **"**The reported error was within 20% for the calculated concentrations against the observations in Beijing (Lu et al., 2019)."

*4) Page 4, Line 11:*
*The SPR analysis (section 4.5) is not meaningful. It is hard to quantitatively estimate the survival fraction of new particles from this equation because the SPR values can be greater than 100% in many cases (Table 1). I think the authors may be able to calculate the loss rate of new particles during each NPF event from CS.*

**Response:** We agree that it does not make sense to calculate the SP beyond 100% because of highly spatial-heterogeneity of NPF in those particular events. In the revision (page 6, lines 4-6), we added "Note that the observed number concentrations of newly grown particles with a larger size sometimes exceeded those with a smaller size under the condition of spatial heterogeneity of NPF. In these cases, that is, NPF events occurring on June 23, and August 12 and 15, SP was not calculated."

The calculated SP is still, however, valid for most of NPF events, which were only slightly affected by the spatial heterogeneity. We rewrote the section 4.5 in the revision.

We argue that the calculated loss rate of new particles would underestimate the contribution of new particles to the population of CCN by even more against the calculated SP. Supposed that NPF mainly occurred at the upper boundary layer, residual layer or upper free troposphere, these grown new particles were mixed down and detected at the building roof site. In contrast, primarily emitted particles were mainly derived from sources at the lower boundary layer. When the particles in different sizes measured at the roof site were used to calculate the loss rate of new particles, the calculated values should be treated as the maximum loss. The loss rate of new particles at loft would be smaller than the calculated value, but the grown new particles at loft rather than at the roof level would act a potential source of CCN. The calculated SP from the observations may be also affected by the increasing loadings of particles when grown new particles mixed down. However, the extent should be smaller than the calculated loss.

Ideally, the vertical profiles of particle number size distributions would be the best to estimate the SP. However, it is practically difficult to obtain the data.

*5) Page 4, Line 13:*
*Please clarify why 3 sigma was chosen.*

**Response:** The particle number concentration follows the lognormal distribution. In the function curve, 1 sigma covers 68% area, 2 sigma covers 95% area, and 3 sigma covers 99% area. In this study, we use 3 sigma to represent almost all particles in this mode. In the revision, we add "3σ covering 99% of the mode particles". In fact, 3 sigma is quietly common approach used in various studies.

*6) Page 4, Line 20: Mass$_{required}$*
*The authors compared Mass$_{required}$ with the changes in mass concentrations of organic and nitrate aerosols, but the latter is generally controlled by accumulation mode particles, not nucleation mode particles. The comparison between Mass$_{required}$ (changes in aerosol mass for nucleation or Aitken mode particles) and the changes in mass concentrations of organic and nitrate aerosols (mainly controlled by accumulation mode particles) is therefore not so meaningful (in sections 4.1-4.3).*

**Response:** In the revision (Page 10, lines 13-24), we added the logic flow in data analysis. It reads as "As mentioned above, the growth of newly formed particles is mainly attributed to sulfuric acid, ammonium nitrate, and secondary organic compounds (Wiedensohler et al., 2009; Riipinen et al.,

2011; Zhang et al., 2012; Ehn et al., 2014; Man et al., 2015; Wang et al., 2015; Burkart et al., 2017; Lee et al., 2019; Wang et al., 2020). We therefore explore their respective contributions as follows. First, we calculated the contribution of sulfuric acid to the growth based on the observed mixing ratio of $SO_2$ and Equations 2–4. Second, we examined whether $NH_4NO_3$ freshly formed in $PM_{1.0}$ or $PM_{2.5}$ during the particle growth period. In case of no $NH_4NO_3$ formation, its contribution would not be expected. This is because an even higher product of $HNO_{3gas}*NH_{3gas}$ is required to overcome the kelvin effect and form $NH_4NO_3$ in nucleation mode and Aitken mode particles. Thus, the growth unexplained by sulfuric acid should be mainly contributed by SOA. Third, in case of $NH_4NO_3$ formation, we compared the net increase in $NH_4NO_3$ with that in SOA. It is noteworthy that this approach is limited by the uncertainty in explaining the growth because the ratios of increased $NH_4NO_3$ over increased SOA in $PM_{1.0}$ or $PM_{2.5}$ may not be the same as the ratios in nucleation mode and Aitken mode particles. In this case, the required mass of $NH_4NO_3$ or SOA to the growth was also estimated and compared with their respective net increases to facilitate the analysis."

In absence of accurate concentrations of chemical compounds in nucleation mode and Aitken mode particles, the above-mentioned approach is one of most reasonable ways to study the growth of newly formed particles. It is an urgent task to accurately measured concentrations of chemicals in those smaller nanometer particles. Unfortunately, no such technologies are commercially available in research community so far.

*7) Page 5, Lines 1-2:*
*Please provide some brief descriptions on model setups.*

**Response:** In the revision (Page 6, lines 21-26), we added "The initial and boundary conditions were obtained from the National Center for Environmental Prediction (NCEP) FNL (Final) Operational Global Analysis datasets (http://rda.ucar.edu/datasets/ds083.2). The major physics options included the Lin microphysics scheme, RRTM long-wave radiation scheme, Goddard short wave scheme, Monin-Obukhov surface-layer scheme, thermal diffusion land-surface scheme, and YSU land-surface scheme. The WRF hourly output files were processed using the Meteorology-Chemistry Interface Processor (MCIP v4.3)."

*8) Pages 5, Lines 8-9:*
*Please describe on model evaluations more clearly (e.g., the degree of agreement with observations, chemical species evaluated).*

**Response:** In the revision (Page 7, lines 4-11), we added "During the study period, the model results generally met the benchmark criteria of the above four species (US-EPA, 2007), with correlations between modeled and measured values larger than 0.57 (Table S1). The modeled concentrations of

$NH_4^+$ reasonably agree with the observations with a normalized mean bias (NMB) of 6%. The NMB slightly increased up to 12% for the modeled concentrations of $SO_4^{2-}$. The modeled values of $NO_3^-$ and SOA were underestimated with NMBs of -29% and -39%, respectively. Underestimation of SOA is a common weakness of the model simulation because a fraction of SOA precursors are not included, and some key formation pathways of SOA may still be missing in current air quality models (Baek et al., 2011; Knote et al., 2014). Detailed evaluation results of this study are provided in the Supporting Information."

*9) Page 5, Lines 23-27:*
*The unit of number concentrations in this paragraph is probably not correct.*

**Response:** Thanks and correct.

*10) Page 6, Lines 2-19:*
*Please clarify why Class II was subclassified to 4 scenarios. What is the purpose of this?*

**Response:** In the revision (Page 8, lines 11-14), we added " The growth of newly formed particles seemingly encountered a ceiling in Scenario 1, in which new particles grown at the maximum $D_{pg}$ unlikely contributed to CCN at normal SS. The ceiling prevented newly formed particles from growing to the CCN size in Scenario 1. The possibility of new particles to grow to CCN size in Scenarios 2–4 remains unknown."

In four scenarios of Type II, Scenario 1 showed a clear evidence, i.e., there was a ceiling existing in the growth of new particles below 50 nm. This is important for modeling and lab studies to explore what cause the ceiling. There was no clear evidences in Scenarios 2-4, where newly formed particles can grow to CCN size. Of course, there was also no clear evidence existing the ceiling.

*11) Page 8, Line 3: the contribution of <2%*
*Please clarify how the authors estimated this contribution. I think the authors have sulfate data observed by AMS. The data can be shown like OM and nitrate in Figures 2-4.*

**Response:** In the revision (Page 5, lines 15-22), we added the equation to calculate the contribution. Please see the revised part.

We believe that the contribution of sulfuric acid to the growth of newly formed particles was much more accurate than the use of the sulfate data observed by AMS. The reason is same as the reviewer's comments in 6).

*12) Page 8, Line 7: 13 ug m-3*

*Please clarify how the authors estimated this value. Did the authors consider the spread of particle size distributions? (like 3 sigma in equation (3)).*

**Response:** In the revision (Page 6, lines 7-14), we added "The amount of chemical species required to grow new particles from $D_{pg1}$ to $D_{pg2}$ ($Mass_{requried}$) is approximately estimated as follows:

$$Mass_{requried} = 4/3\pi \, [(D_{pg2}/2)^3 - (D_{pg1}/2)^3] * N * \rho \tag{5}$$

$\rho$ is the density, which is assumed as 1.5 µg m$^{-3}$ for OOA (or SOA) and 1.7 µg m$^{-3}$ for $NH_4NO_3$, respectively. Considering that the particle number concentration may decrease because of the dry deposition, diffusion and dilution effects, and particle coagulation, N represents the integral value of new particle number concentrations with the geometric median diameter of new particles from $D_{pg2-3\sigma}$ to $D_{pg2+3\sigma}$. The approximate value may overestimate the required amount because particle–particle coagulation has not been deducted."

*13) Page 8, Lines 6-8:*

*As I described above, the comparison between the required mass (13 ug m$^{-3}$) and PM$_{1.0}$ enhancement (15 ug m$^{-3}$) is not so meaningful because the former focuses on nucleation/Aitken mode particles but the latter is usually dominated by accumulation mode particles.*
*I think what the authors can do here is to calculate mass concentration changes for sulfate, nitrate, ammonium, and SOA and to discuss which changes are the largest during the growth periods of new particles.*

**Response:** In the revision (Page 10, lines 13-24), we added the logic flow in data analysis. Please see our response to comment 6). Using the logic flow, we revised our analysis accordingly through the manuscript. Please see the revision.

*14) Page 8, Line 7:*

*OM can be divided into HOA (POA like) and OOA (SOA like) by using m44 and m57 signals. Only OOA can contribute to the growth of particles.*

**Response:** Agree. In the revised manuscript, we use the OOA data instead of OM to discuss the growth of new particles. Please see the revision.

*15) Page 9, Lines 3-4:*

*I don't agree with this authors' description. The simulated OA and nitrate cannot be used to interpret the data unless the authors evaluate the simulations with observations.*

**Response:** See our responses to comment 8).

*16) Page 10, Line 16 Delete "(ON)".*

**Response:** Corrected.

*17) Page 10, Line 28: OM can be divided to HOA and OOA as I described above.*

**Response:** We will change OM to OOA in the main text and figures.

*18) Page 11, Lines 5-6: "Repartition of the…" This part should be removed because no data can support this sentence.*

**Response:** In the revision (Page 13, lines 19-22), it now reads as **"**During this period of shrinkage, the observed mixing ratio of $O_x$ largely decreased from 130 ppb to 80 ppb, and the observed OOA decreased from 16.2 μg m$^{-3}$ to 11.4 μg m$^{-3}$ (Fig. 4b, c). However, the concentrations of $NH_4^+$ were almost constant. Repartition of the semivolatile SOA in gas and particle phases was hypothesized to cause the evaporation of semivolatile SOA to the gas phase.**"**

*19) Page 12, Line 19 "then in", "transience": they should be corrected.*

**Response:** Corrected.

*20) Page 12, Line 26: Section 4.5 As I described above, this section is not so meaningful and should be removed. How did the author consider the contribution of primary particles in this analysis?*

**Response:** Please see our comment 4) and more information presented below:
We used a high-time resolution PARALLELING particle sizer, i.e., FMPS, to measure the particle number size distribution (PNSD) in 1 s time resolution. The high time resolution of FMPS can allow clearly identify the signals of newly formed particles from preexisting ambient particles, e.g., freshly emitted particles from combustion, as well as the mixing process of the different types of particles (Liu et al., 2014; Man et al., 2015; Zhu et al., 2017, 2019).

According to our previous studies (e.g. Zhu et al., 2017, ACP) and the review paper by Tuan et al. (2015), the PNSD from traffic emissions are characterized by two peaks, i.e., about 16 nm and 30 nm, and intermittently last a few seconds or minutes (Figure R1). Figure R2 (added in new supporting information) shows the fresh industrial emissions associated with $SO_2$ and/or the cooking emissions associated with increased COA at 18:00-20:00. It is clear that the dominant modes of

particles from traffic emissions, industrial emissions and cooking emissions occurred at ~20 nm, ~30 nm and ~40 nm, respectively, and the domain mode size were quietly stable in the study period. Their contributions were important to the particle number concentrations during non-NPF periods, which were not the focus of this study. On the roof sampling site, their contributions to the observed particle concentrations during the initial few hours of NPF were generally negligible in presence of wind speeds of 4-6 m s$^{-1}$, except a few occasional spikes lasting in minutes. These spikes were excluded in calculating NMINP. In the revision, we add "Note that a few spikes of several minutes were occasionally observed and were excluded to calculate NMINP because they may reflect primary particles from localized sources (Liu et al., 2014; Zhu et al., 2017)." The same is true in calculating the growth rate of particles, shrinkage rate and $D_{pgmax}$. Additionally, their influences can also be ignored in studying the growth of newly formed particles when the particles grew over 50 nm.

During the NPF events (e.g. Figure R3), COA occasionally influences the new particles signal, and the growth of new particles was consistent with the increase in LO-OOA and MO-OOA. Therefore, we argued that the growth of new particles depends largely on the condensational growth.

We prefer to add the argument in Supporting Information since the challenge is not immediately related to the samplings and associated analyses presented in the text.

[Figure]

Figure R1 Contour plot of traffic emissions (a) and the raw FMPS data of the vehicle spikes (b).

[Figure]

Figure R2 Fresh industrial emissions with high SO$_2$ (12:30-15:00) and the cooking emissions with increased cooking OA (COA, 18:20-21:00).

[Figure]

Figure R3 NPF event and the variations of hydrocarbon-like OA (HOA), cooking OA (COA), less oxidized oxygenated OA (LO-OOA) and more oxidized oxygenated OA (MO-OOA) on June 18, 2014.

**Reference:**

[revised manuscript text omitted]

---

## Author Response (AR2)

**Response to reviewer's comments**

*Editor Decision: Publish subject to minor revisions (review by editor) (22 Oct 2020) by Stefania Gilardoni*
*Comments to the Author:*
*One of the reviewers suggested further revision of the manuscript. Here is the list of the changes suggested by the reviewer.*

*1) Page 2, line 24-*
*The authors summarized previous NPF studies in Beijing well in this paragraph. However, it is not clear yet in the revised manuscript what are different from previous studies and what are scientifically new. For example, in Page 3, lines 3-6, the authors described as follows: "sulfuric acid and/or organic vapors have been proposed to drive particle growth in different NPF events (Wiedensohler et al., 2009; Yue et al., 2010; Wu et al., 2016). Recently, the formation of $NH_4NO_3$ has been proposed as a driver of the rapid growth of newly formed particles in field studies and chamber experiments (Zhu et al., 2104; Man et al., 2015; Wang et al., 2020)." Which parts of this study are different from these previous studies? How was the understanding of NPF in Beijing advanced in this study? Because the importance of $NH_4NO_3$ and organics has already been found in previous studies, the sentence at Page 3, lines 6-7 ("The role of $NH_4NO_3$ in the growth of newly formed particles in Beijing remains poorly understood.") is not enough.*

**Response:** In the revision, page 3, the top paragraph, after "The patterns of particle growth have not been well characterized.", we add "For example, the occurrence frequency of NPF events in which newly formed particles can grow to CCN size is virtually unknown. The same can be said to season trends in the occurrence frequency.". After "The role of $NH_4NO_3$ in the growth of newly formed particles in Beijing remains poorly understood." we add "The contributions of sulfuric acid, organic vapors and $NH_4NO_3$ to the growth of newly formed particles at different sizes are also non-existent."

*2) Page 5, lines 23-*
*The SP analysis (this part and section 4.5) should be removed. I cannot understand why the authors can say that the calculated SP is valid for most of NPF events. This analysis implicitly assumes that all atmospheric particles in Beijing are formed during a new particle formation event in Beijing. I think this assumption is unrealistic. Some aerosol particles in Beijing may be preexisting particles (e.g., primary particles, new particles formed a few days ago), transported from other cities, and/or transported from higher altitudes. Although fresh primary particles may be small enough, many particles in the atmosphere (>50 nm or >70 nm) are probably produced by non-NPF or non-local NPF processes. This analysis is therefore not valid and should be excluded from the manuscript.*

**Response:** We thank very much for the reviewer fastening upon this point. This helps us to eventually find the right way to calculate the net maximum number concentration of grown new particles beyond 50 nm or 70 nm. The same can be said to the calculated SP. In the revision, page 5, we revised "To calculate the survival probability (SP) of grown new particles beyond 50 nm or 70 nm, temporal variations in $N_{50-200 \text{ nm}}$ and $N_{70-200 \text{ nm}}$ were plotted, e.g., Fig. S2a-b showed the temporal results on 25 August 2014. The stable minimum $N_{50-200 \text{ nm}}$ or $N_{70-200 \text{ nm}}$ can be clearly identified approximately 2-3 hours after the NPF event to be observed. The stable maximum $N_{50-200 \text{ nm}}$ or $N_{70-200 \text{ nm}}$ can be also clearly identified approximately 11-13 hours later. The difference between the two values was used to estimate the net maximum number concentration of grown new particles beyond 50 nm or 70 nm, i.e., $N_{50-200 \text{ nm}}$ (*net*) and $N_{70-200 \text{ nm}}$ (*net*). Thus, $SP_{50 \text{ nm}}$ and $SP_{70 \text{ nm}}$ were estimated as:

$$SP_{50 \text{ nm}} = \frac{N_{50-200 \text{ nm}} \, (net)}{\text{NMINP}} \times 100\% \quad (5)$$

$$SP_{70 \text{ nm}} = \frac{N_{70-200 \text{ nm}} \, (net)}{\text{NMINP}} \times 100\% \quad (6)$$

[Figure]

Fig. S2 Schematic diagram of $N_{50-200 \text{ nm}}$ (*net*) (a) and $N_{70-200 \text{ nm}}$ (*net*) (b) on 25 August 2014"

Using the right approach, the calculated $SP_{50 \text{ nm}}$ and $SP_{70 \text{ nm}}$ have been updated in the revision. So does the analysis.

*3) Page 6, line 9, equation (5)*
*Please clarify why $D_{pg1}$ and $D_{pg2}$ (median diameters) were used instead of observed particle diameters. It is more straightforward to use the observed diameters of the size distribution observations directly rather than using the $D_{pg1}$ and $D_{pg2}$.*

**Response:** In the revision, page 6, line 4, we add "Followed Equation (4) proposed by Kulmala et al., (2001), in which $D_{pg1}$ and $D_{pg2}$ are used to calculate C,".

*4) Page 6, lines 16-, Section 2.3*

*The model simulations cannot be used to interpret the relative importance of NH₄NO₃ and SOA because SOA is severely underestimated. Model simulations and their interpretation should be removed from the manuscript. The main results of this study do not change even if the model results were excluded. I think the discussion using the model simulations decreases the reliability of the whole results shown in this manuscript.*

**Response:** We agree the model simulations cannot be used to interpret the relative importance of $NH_4NO_3$ and SOA. However, the modeled results can be used to argue the importance of $NH_4NO_3$ based on the reasonable performance in comparison with the observations. In the revision, quantitative analysis of simulated ammonium nitrate and semi-quantitative analysis of simulated SOA were used. Please see the revision.

*5) Page 7, line 8*

*NMB of -39% is probably wrong for SOA when I see Figure S7. Simulated SOA is severely (more than an order of magnitude?) underestimated.*

**Response:** The NMB was mistakenly calculated by using the simulated SOA against the measured OOA. The mistake has been corrected in the revised Table S1. This does not affect the Fig. S7. Sorry for this error and thank very much for the important comments.

*6) Page 7, line 15*

*Figure S2 can be used in the main manuscript.*

**Response:** Agree. Done.

*7) Page 8, line 8-*

*Line 10: Fig S3b -> S3c, line 16: Fig S3c -> S3e, line 18: Fig S3d -> S3g, line 20 Fig. S3e -> S3i*

**Response:** Done.

*8) Page 10, line 26*

*Fig 2b does not show that sulfuric acid contribution is small.*

**Response:** The sentence has been revised as "Based on the observed mixing ratio of $SO_2$ shown in Fig. 3b and Equations 2–4, sulfuric acid was estimated to contribute < 2% to particle growth during the whole NPF period"

*9) Page 11, line 19-Page 12, line 5*

*These three paragraphs should be removed as I described in the comment 4.*

**Response:** See our response above.

*10) Page 13, lines 5-8*

*This paragraph should be removed also.*

**Response:** See our response above.

*11) Page 14, lines 4-12*

*Authors explain what was obtained but do not explain why this result was obtained.*

**Response:** The sentences have been revised as "The growth rate of newly formed particles is an intensity quantity and mainly determined by the concentrations of condensable vapors such as sulfuric acid, organics of various volatilities, nitric acid, and ammonia (Zhang et al., 2012; Ehn et al., 2014; Man et al., 2015; Lee et al., 2019). In contrast, $D_{pgmax}$ values are determined by the total amount of vapors condensed on grown new particles (an extensive quantity), which may or may not have a positive correlation with the concentrations of these vapors (Zhu et al., 2019)."

*12) Section 4.5*

*This section should be removed as I commented before.*

**Response:** See our response above.

---

## Author Response (AR3)

Dear Stefania,

Thank you help for handing the peer-review of this work. Your comments, our response and the revision are listed below:

***Comments:*** *When you calculate the survival probability (SP), you are implicitly assuming that the increment of N50-200 and N70-200 are not affected by the boundary layer height (in Fig S2 the N increment overlaps with the decreasing boundary layer height). I would suggest to the authors to comment this point and specify if the SP calculated using such increment is actually an upper bound.*

**Response:** In the revision, page 5, lines 24-30, we added "With increasing of the height of planetary boundary layer (PBL, Fig. S2d), the stable minimum $N_{50-200 \text{ nm}}$ or $N_{70-200 \text{ nm}}$ can be clearly identified approximately 2-3 hours after the NPF event to be observed. The minimum $N_{50-200 \text{ nm}}$ ($N_{70-200 \text{ nm}}$) remained constant for approximately two (four) hours, even though the height of PBL increased continuously. The new particle signal likely mixed well within the whole PBL. The stable maximum $N_{50-200 \text{ nm}}$ or $N_{70-200 \text{ nm}}$ can be also clearly identified approximately 11-13 hours later, when the height of PBL has been lowered down a lot from the maximum. The change in height of PBL had no detectable influence on the maximum $N_{50-200 \text{ nm}}$ or $N_{70-200 \text{ nm}}$ because the growth of >30 nm new particles to a larger size probably occurred within the PBL."

In addition, in Supporting Information, "The height of planetary boundary layer at the nearby site was obtained from the ECMWF reanalysis data, which was downloaded from https://cds.climate.copernicus.eu/cdsapp#!/dataset/reanalysis-era5-single-levels?tab=overview."
has been added in Fig. S2d.

We hope that the revision can help the readers.

Best Regards,

Xiaohong

Prof. Xiaohong Yao (Ph.D)
Ocean University of China